# Optimizing ID Consistency in Multimodal Large Models: Facial Restoration via Alignment, Entanglement, and Disentanglement

**Yuran Dong, Hang Dai**[*]**, Mang Ye**[*]

National Engineering Research Center for Multimedia Software
School of Computer Science, Wuhan University, Wuhan, China
{dongyuran, daihang, yemang}@whu.edu.cn

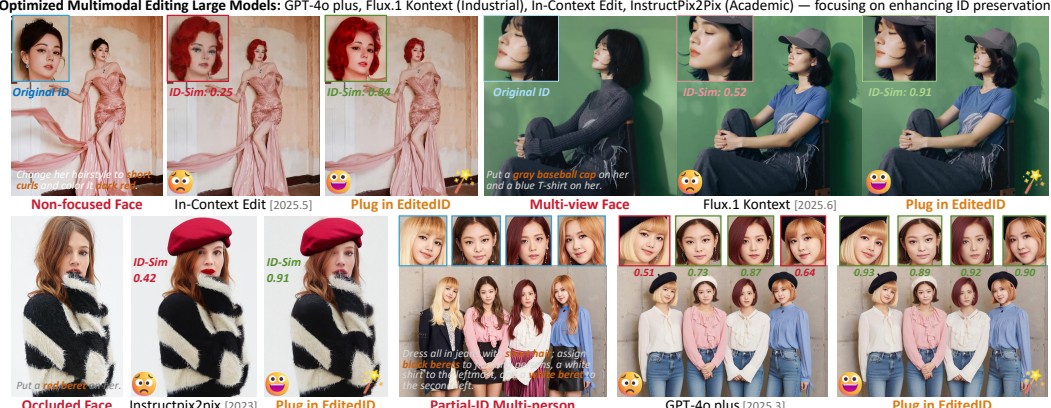

Figure 1: EditedID Optimized Multimodal Editing Large Models (Academic/Industrial): ID consistency capability. ID-Sim (same ID > 0.7, green border; otherwise red) indicates identity preservation in challenging scenarios: Non-focused, Multi-view, Occluded Faces, Partial-ID Multi-persons.

## Abstract

Multimodal editing large models have demonstrated powerful editing capabilities across diverse tasks. However, a persistent and long-standing limitation is the decline in facial identity (ID) consistency during realistic portrait editing. Due to the human eye's high sensitivity to facial features, such inconsistency significantly hinders the practical deployment of these models. Current facial ID preservation methods struggle to achieve consistent restoration of both facial identity and edited element IP due to *Cross-source Distribution Bias* and *Cross-source Feature Contamination*. To address these issues, we propose **EditedID**, an Alignment-Disentanglement-Entanglement framework for robust identity-specific facial restoration. By systematically analyzing diffusion trajectories, sampler behaviors, and attention properties, we introduce three key components: 1) Adaptive mixing strategy that aligns cross-source latent representations throughout the diffusion process. 2) Hybrid solver that disentangles source-specific identity attributes and details. 3) Attentional gating mechanism that selectively entangles visual elements. Extensive experiments show that EditedID achieves state-of-the-art performance in preserving original facial ID and edited element IP consistency. As a training-free and plug-and-play solution, it establishes a new benchmark for practical and reliable single/multi-person facial identity restoration in open-world settings, paving the way for the deployment of multimodal editing large models in real-person editing scenarios. The code is available at https://github.com/NDYBSNDY/EditedID.

---

[*]Corresponding authors.

# 1 INTRODUCTION

Recent years have witnessed growing interest in Multimodal Editing Large Models Zhang et al. (2025); Liu et al. (2025); Black Forest Labs (2025); OpenAI (2025); Deng et al. (2025); Wu et al. (2025) owing to their broad applicability and practical value. These models function by interpreting user instructions to enable image editing. Although these models are effective for identity preservation in cartoons, they degrade significantly with complex long-instruction real-person fashion editing. Detailed prompts like "make him wear a light gray jacket with black-framed glasses" frequently induce facial artifacts. Academic models, such as In-ContextEdit Zhang et al. (2025) and Instruct-Pix2Pix Brooks et al. (2023), suffer from limited fine-tuning data, affecting facial feature extraction and causing progressive deterioration of facial edits with long instructions. Industrial models, such as GPT-4o Plus OpenAI (2025), Qwen-Image-Edit Wu et al. (2025) and Flux.1 Kontext Black Forest Labs (2025) prioritize LLM-driven textual controllability but neglect facial geometric constraints, producing random facial identities. Given the high sensitivity of human perception to facial features, even slight deviations in identity can render the results unusable. However, due to the confidentiality of real-world facial datasets, task-specific fine-tuning is often impractical. Multimodal editing large models face the long-standing challenge of achieving reliable identity preservation when performing complex, long-instruction edits Dong & Ye (2025); Wang & Ye (2024) in real-person scenarios.

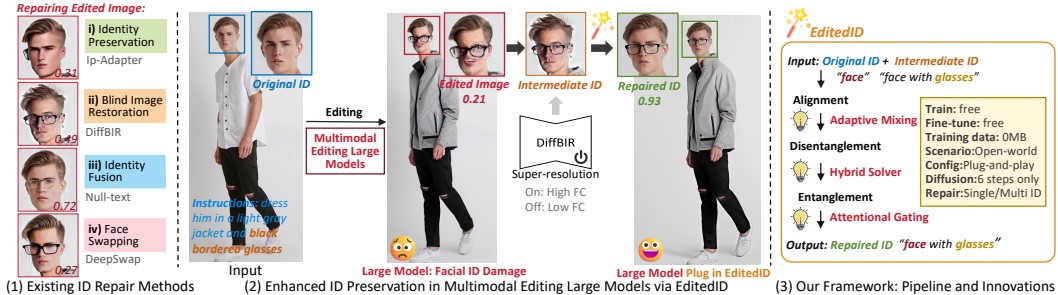

Figure 2: Enhanced by EditedID: ID preservation in Multimodal Editing Large Models. Comparison of repair effects and innovations. DiffBIR pre-repair triggered by Facial Corruption (FC) severity.

To optimize the ID consistency capability of existing multimodal editing large models, as shown in Fig. 2 (2), our goal is to achieve plug-and-play facial identity reconstruction. This involves restoring facial consistent with the Original ID while preserving edited Element IP (e.g., the color, size, or texture of accessories such as glasses) consistent with the Intermediate ID. In Fig. 2 (1), existing four types of facial consistency methods each exhibit limitations in achieving this objective:

**Cross-source Distribution Bias Leads to:** *i) Detail Loss in Identity Preservation.* Identity-preserving approaches Ye et al. (2025); Ohayon et al. (2024); Ye et al. (2023a) often degrade photorealism by blurring fine facial details and introducing cartoon-like artifacts. This degradation primarily stems from Cross-source Distribution Bias. For instance, IP-Adapter Ye et al. (2023a) merges coarse-grained identity features (learned from limited data) with high-resolution features from base diffusion models, leading to distribution mismatches and distortion artifacts. *ii) Original ID Lost in Blind Restoration.* Generic facial restoration methods Hu et al. (2025); Ito et al. (2025); Lin et al. (2024) focus on facial super-resolution but neglect identity consistency. Although these models generate clear textures and plausible structures, Cross-source Distribution Bias between the generated facial features and the original distribution leads to random, non-ID-specific facial reconstruction.

**Cross-source Feature Contamination Leads to:** *iii) Element IP Loss in Identity Fusion.* Fusion-based methods Yao et al. (2025); Nam et al. (2024); Chen et al. (2024); Ye et al. (2023b) act as a relaxed feature transfer mechanism. During cross-source feature (original facial and edited element) fusion, such methods often suffer from inter-feature contamination, leading to loss of fine-grained edited attributes. For instance, the "black-framed" style of glasses is frequently lost, as these methods prioritize element-level consistency over accurate element attribute preservation. *iv) Edited Facial Noise in Face-Swapping.* Face-swapping methods Ye et al. (2025); Wang et al. (2025); Wang (2024); Liu et al. (2024); Li et al. (2024) exhibit high sensitivity to artifacts in edited facial. Effective under normal facial conditions, their performance degrades significantly in the presence

of geometric or structural distortions, where Cross-source Feature Contamination (original ID and edited facial) and leads to loss of the original ID in swapping.

To address these limitations, we draw inspiration from 3D facial processing methods Li et al. (2023), which separate specific elements (e.g., glasses and faces) from cross-source objects and re-simulate physical interactions (e.g., lighting, position) to combine them into novel composites. This inspires the core principle of our proposed **EditedID** (Fig. 2 (2) and (3)) in 2D scenarios: **Alignment–Disentanglement–Entanglement**. EditedID is a diffusion-inversion-based ID-consistent facial reconstruction method. Specifically, to mitigate *Cross-source Distribution Bias*, we introduce Adaptive Mixing for dual-ID (Original and Intermediate ID) latent Alignment; to isolate *Cross-source Feature Contamination* in aligned identities and preserve personalized element features, we propose a Hybrid Solver for dual-ID latent Disentanglement. Finally, via an Attentional Gating mechanism, we Entangle facial attributes from the Original ID with edited elements from the Intermediate ID. Refer to Appendix A.1 for related work. Our contributions are as follows:

❶ **Trajectory Insight — Adaptive Mixing:** By revealing the multi-solution and controllability of diffusion trajectories, we propose Adaptive Mixing—a cross-object feature fusion approach that mitigates *Cross-source Distribution Bias* to avoid abrupt transitions and artifacts.

❷ **Sampler Analysis – Hybrid Solver:** By leveraging DDIM identity retention and DPM-Solver++ detail enhancement properties, we design Hybrid Solver—a global-timestep hybrid sampling method that isolates *Cross-source Feature Contamination* while preserving original identity-detail features.

❸ **Attention Mechanism – Attentional Gating:** By uncovering the distinct roles of attention in diffusion processes, we introduce Attentional Gating—an specified element control mechanism that preserves single-element structures and balances multi-element interactions during entanglement.

❹ **Identity-Consistent Restoration Paradigm:** We propose a novel, training-free trajectory Alignment framework for real-world single- and multi-facial restoration, leveraging the Disentanglement and Entanglement of cross-source semantic-specific elements within pre-trained diffusion models.

## 2 METHODOLOGY

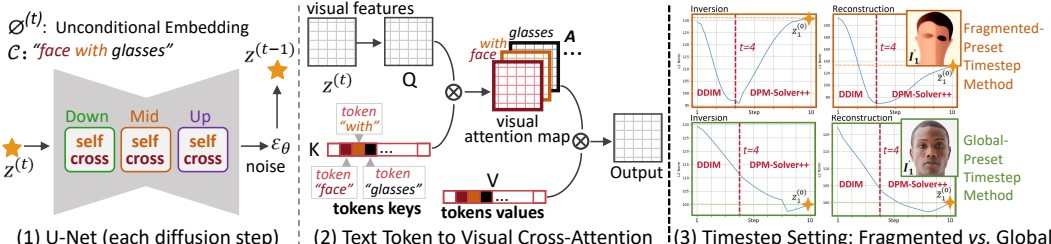

Figure 3: **Preliminaries:** (1) U-Net (2) cross-attention. **Hybrid solver:** (3) Timestep Init. (Sec.2.3).

**Preliminaries.** Diffusion models synthesize an image $z^{(0)}$ by iteratively denoising random noise $z^{(T)}$ over $T$ steps. The inversion process diffuses a real image $z^{(0)}$ to noise $z^{(T)}$, while the reconstruction path reverses it via a denoising trajectory. Null-text optimization Mokady et al. (2023) bridges the deviation between reconstruction and inversion by introducing an unconditional embedding $\emptyset^{(t)}$, achieving accurate reconstruction trajectories of real images $\bar{z}^{(0)} \approx z^{(0)}$. This process is training-free. At each step $t$, U-Net predicts noise $\varepsilon = \varepsilon_\theta(z^{(t)}, t)$ using $z^{(t)}$, conditional embedding $\mathcal{C}$, and $\emptyset^{(t)}$; the sampler then updates $z^{(t-1)}$ accordingly in Fig. 3(1).

**Diffusion Samplers:** DDPM Ho et al. (2020) adopts a Markov chain (e.g., $T = 1000$), incurring high cost. DDIM Song et al. (2020) accelerates sampling via non-Markovian steps (e.g., $T = 50$). It reconstructs $\tilde{z}^{(0)}$ from $z^{(t)}$ and deterministically computes $z^{(k)}$, $k < t$. DPM-Solver++ Lu et al. (2022) treats reverse diffusion as an ODE, solved via high-order Taylor approximations. This enables rapid convergence with realistic details (details in Appendix A.2).

**U-Net Attention:** U-Net includes cross-attention layers (Fig. 3(2)) that map text tokens to visual attention maps $A$. Query $Q$ is projected from image features $z^{(t)}$, while Key $K$ and Value $V$ from

$\mathcal{C}$. Prompt-to-Prompt Hertz et al. (2022) manipulates attention maps for feature transfer. Null-text extends this to real-image editing via reconstruction trajectory.

**Problem Statement:** Given a source identity $I_1$ and an edited image $I_2$ (e.g., with black glasses), the goal is to reconstruct a new image $I_3$ that preserves $I_1$'s identity while retaining $I_2$'s semantics. For degraded $I_2$, we first apply DiffBIR Lin et al. (2024) for super-resolution enhancement.

## 2.1 MOTIVATION

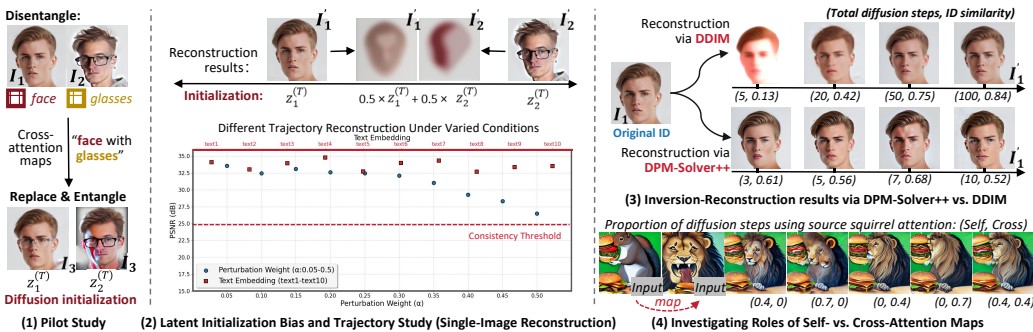

Figure 4: (1) Limitations of the Pilot Study and analysis of three contributing factors: (2) (3) (4).

**Pilot Study.** We investigate Prompt-to-Prompt and Null-text for identity-consistent restoration of edited images, focusing on feature disentanglement and fusion via cross-attention. In Fig. 4 (1), ***Latent Initialization*** of $I_3$ is achieved by using either $z_1^{(T)}$ or $z_2^{(T)}$, which are obtained through ***DDIM Sampler*** inversion of *Dual-ID* ($I_1$ and $I_2$). During the diffusion process of $I_3$, ***Cross-Attention*** maps ($I_1 \rightarrow$"face", $I_2 \rightarrow$"glasses") are fused to guide cross-source feature transfer. However, $I_3$ exhibits asymmetric feature retention—starting from $z_1^{(T)}$ preserves ID (face) but loses edited attributes (glasses), and vice versa. Significant artifacts also emerge in $I_3$. Key factors include:

**1) Latent Initialization Bias:** Different initial latents ($z_1^{(T)}$ *vs.* $z_2^{(T)}$) produce incompatible attention maps, leading to poor fusion. Mixing latents ($0.5z_1^{(T)} + 0.5z_2^{(T)}$) amplifies artifacts due to diffusion nonlinearity (Fig. 4 (2)-top). To reduce cross-source latent space bias, we explore the controllability of diffusion trajectories. In Fig. 4 (2)-bottom, trajectories are perturbed via text embedding modifications or noise injection ($\alpha \in [0.05, 0.5]$). Higher PSNR Fardo et al. (2016) indicates to greater reconstruction-original similarity ($\bar{z}^{(0)} \approx z^{(0)}$), consistency threshold: 25 dB. Similar PSNR under different perturbations suggest identical reconstructions across trajectories. Our reveal key findings:

**Observation 1 — Trajectory: i) Multi-solution.** Multiple trajectories can achieve same output (same PSNR) if fixed target $z^{(0)}$, demonstrating multi-path solvability ($\alpha = 0.25$ or $text = text5$) of $z^{(0)}$. **ii) Controllability.** $z^{(T)}$ and evolution jointly shape trajectories; controlled perturbations ($\alpha$) allow trajectory alteration without degrading fidelity (PSNR $> 25$ dB).

**2) DDIM Sampler Loss Detail:** DDIM sampler introduces bias in detail fidelity, which is exacerbated by cross-source fusion and leads to artifacts. Fig. 4(3) compares ID similarity Deng et al. (2019) ($> 0.7$ same ID) between DPM-Solver++ and DDIM under different reconstruction steps.

**Observation 2 — Sampler: i) DDIM: Identity over Details.** More diffusion steps ($> 50$) with deterministic path preserves identity consistency (ID sim: 0.84), but detail loss due to 1st-order smoothing and error accumulation. **ii) DPM-Solver++: Details over Identity.** Fewer diffusion steps ($< 10$) with high-order Taylor expansion produce high-fidelity details, yet path deviation and prior interference cause identity loss (Avg. ID sim: 0.59). Detailed analysis in Appendix A.3.

**3) Cross-Attention Weak Constraints:** Cross-source feature transfer via cross-attention alone yields weak constraints and ignores spatial coupling in self-attention. We analyze self *vs.* cross roles by manipulating attention maps ("squirrel" replace "lion") under varied strengths in Fig. 4 (4):

**Observation 3 — Attention: i) Self: Single-Element Structure.** Self-attention encodes single-element structure; increasing self-strength enforces shape transfer (lion$\rightarrow$squirrel). **ii) Cross:**

**Multi-Element Interaction.** Cross-attention encodes multi-element interaction; increasing cross-strength enhances semantic interactions (lion *eating* hamburger). **iii) Balance:** Coordinated cross/self-attention tuning is essential for structure-interaction trade-offs and coherence outcomes.

These observations motivate us to solve three core challenges, solved in Sec. 2.2, Sec. 2.3, Sec. 2.4.

- **C1:** *How to align dual-ID diffusion trajectories to reduce cross-source latent initialization bias?*

- **C2:** *How to integrate DDIM and DPM-Solver++ for identity-detail preserving disentanglement?*

- **C3:** *How to coordinate cross/self-attention for structure-interaction balanced entanglement?*

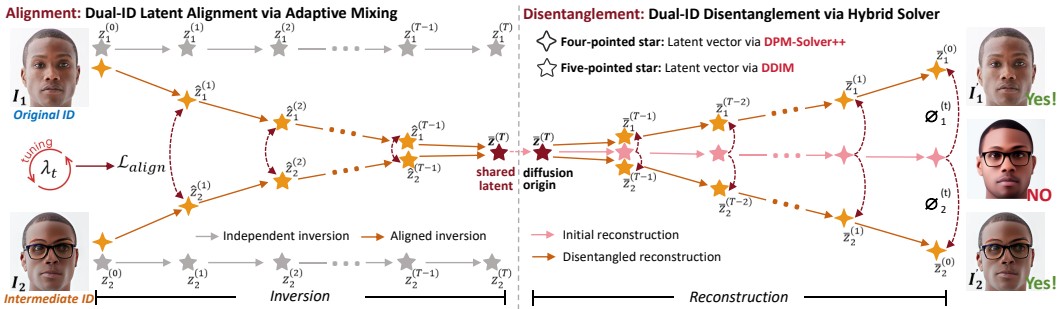

Figure 5: Dual-ID alignment (Left) and disentanglement (Right).

## 2.2 ALIGNMENT: DUAL-ID LATENT ALIGNMENT VIA ADAPTIVE MIXING

Linear blending the invert trajectories of the source images $I_1$ and $I_2$ by combining their latent codes $z_1^{(t)}$ and $z_2^{(t)}$ with *fixed weights* leads to two critical failures: 1) Early inversion (near $z^{(0)}$), excessively abrupt averaging of latents causes significant degradation of source-specific features; 2) Late inversion (near $z^{(T)}$), lack of adaptivity results in overlapping trajectories, leading to cross-source feature contamination. To address **C1** based on **Observation 1**: *Multi-solution and Controllability* of trajectory, we propose Adaptive Mixing with learnable weights $\lambda_t$ (Fig. 5). The $\lambda_t$ dynamically balances the contributions of $\hat{z}_1^{(t)}$ and $\hat{z}_2^{(t)}$, optimized via gradient descent to minimize alignment loss (in red):

$$\mathcal{L}_{align} = \left\| \hat{z}_1^{(t)} - \hat{z}_2^{(t)} \right\|_2^2, \quad \lambda_t \leftarrow \lambda_t - \eta \cdot \bigtriangledown_{\lambda_t} \mathcal{L}_{align}, \tag{1}$$

where $\eta = 0.01$ is the learning rate, $\lambda_t \in [0, 0.5]$. During inversion, latent codes ($\hat{z}_1^{(t)}$ and $\hat{z}_2^{(t)}$) are updated independently (noise predicted separately for $I_1$ and $I_2$ ):

$$\hat{z}_1^{(t+1)} = (1 - \lambda_t) \cdot \hat{z}_1^{(t)} + \lambda_t \cdot \hat{z}_2^{(t)}, \ \hat{z}_2^{(t+1)} = (1 - \lambda_t) \cdot \hat{z}_2^{(t)} + \lambda_t \cdot \hat{z}_1^{(t)}. \tag{2}$$

Lower $\lambda_t$ initial values yield smoother trajectory alignment. To ensure alignment of $\hat{z}_1^{(t)}$ and $\hat{z}_2^{(t)}$ in latent space at $t = T$, we enforce constrained convergence in later diffusion steps:

$$\hat{z}_1^{(t+1)} = \hat{z}_2^{(t+1)} = (\hat{z}_1^{(t)} + \hat{z}_2^{(t)})/2, \quad \lambda_t = 0.5. \tag{3}$$

This yields a unified initialization $\bar{z}^{(T)}$ with two smooth merging paths (free from abrupt loss variations), which preserve original features while harmonizing latent trajectories. By storing disentangled source-specific attributes along the paths, our method achieves latent space alignment, mitigates *Cross-source Distribution Bias*, and retains individuality (e.g., facial identity ($I_1$) and glasses ($I_2$)).

## 2.3 DISENTANGLEMENT: DUAL-ID DISENTANGLEMENT VIA HYBRID SOLVER

After aligning the initial latent space $\bar{z}^{(T)}$ for dual identities, we disentangle original and intermediate ID features from this shared initialization point for selective token fusion. Extending null-text

optimization, we optimize distinct null-text embeddings $\left\{\emptyset_i^{(t)}\right\}_{t=1}^{T}$ per ID to minimize MSE between reconstructed latents $\bar{z}_i^{(t-1)}$ and aligned states $\hat{z}_i^{(t-1)}$:

$$\mathcal{L}_{rec} = \sum_{i=1}^{2} \left\| \hat{z}_i^{(t-1)} - z_{t-1}(\bar{z}_i^{(t)}, \emptyset_i^{(t)}, \mathcal{C}_i) \right\|_2^2. \tag{4}$$

Unlike single image null-text optimization, our reconstruction objective $\mathcal{L}_{rec}$ integrates joint MSE losses for both $I_1$ and $I_2$ reconstructions ($i = 1, 2$), conditioned on embeddings $\mathcal{C}_i$. Exclusive DDIM sampling updates subsequent latents via: $\bar{z}_i^{(t-1)} = z_{t-1}(\bar{z}_i^{(t)}, \emptyset_i^{(t)}, \mathcal{C}_i)$. Under trajectory alignment constraints, this amplifies detail degradation and artifacts in the reconstructed IDs.

To resolve **C2** with **Observation 2**—*DDIM: Identity over Details, DPM-Solver++: Details over Identity*—we propose a Hybrid Solver that dynamically invokes DDIM or DPM-Solver++ during $\bar{z}_i^{(t-1)}$ prediction to harness complementary advantages:

$$\bar{z}_i^{(t-1)} = \begin{cases} DPM\text{-}Solver(\bar{z}_i^{(t)}, \emptyset_i^{(t)}, \mathcal{C}_i), t \in [s_1, s_2], \\ DDIM(\bar{z}_i^{(t)}, \emptyset_i^{(t)}, \mathcal{C}_i), otherwise. \end{cases} \tag{5}$$

where $s_1$ and $s_2$ denote start/end steps for DPM-Solver++ invocation. Empirical analysis (Appendix A.8) reveals optimal reconstruction strategy: DDIM invocation in early steps (near $\bar{z}^{(T)}$) establishes robust ID feature preservation, while DPM-Solver++ activation in the late steps (near $\bar{z}^{(0)}$) actively repairs and enhances the textural details from DDIM-retained features. Our hybrid solver optimizes diffusion timestep sampling, enabling high-fidelity ID reconstruction in few steps while resolving the long-standing efficiency-fidelity trade-off in DDIM sampling. Furthermore, it ensures identity-detail consistent disentanglement for dual-ID, preventing *Cross-source Feature Contamination*.

**Symmetry Constraint:** To ensure feature alignment when using the same sampler at matching noise levels, the invocation of DPM-Solver++ should be symmetric across both the inversion and the reconstruction stages in Fig. 5.

**Timestep Continuity:** During the inference of $\bar{z}_i^{(t-1)}$, the sampler requires a consistent sequence of timesteps corresponding to the full diffusion step length $t = T$. However, different samplers compute their timestep sequences differently (see Appendix A.4). In our initial attempt, we adopt a fragmented scheduling approach, where the diffusion trajectory is partitioned into fixed intervals, with DDIM applied over $[0, s_1)$, DPM-Solver++ over $[s_1, s_2]$, etc.

As shown in Fig. 3(3), employing a 11-step reconstruction schedule (DDIM for steps 0–4; DPM-Solver++ for steps 5–10) under symmetric inversion leads to uncontrolled latent divergence at the transition boundary ($t = 4$). This discontinuity resulted in chromatic aberrations and significant reconstruction errors ($z_1^{(0)} \neq \bar{z}_1^{(0)}$) in Fig. 3(3)-top.

To resolve this, we propose a global timestep pre-setting strategy that ensures temporal continuity across the entire $[0 \rightarrow T]$ trajectory. Specifically, we first pre-compute the full timestep sequences for both schedulers: DDIM: $\{\tau_0, \tau_1, ... \tau_{T-1}\}$, DPM-Solver++: $\{\sigma_0, \sigma_1, ... \sigma_{T-1}\}$. we dynamically assign the timestep at each diffusion step $t$ as:

$$timestep = \begin{cases} \sigma_t, & t \in [s_1, s_2], \\ \tau_t, & \text{otherwise.} \end{cases} \tag{6}$$

As shown in Fig. 3(3)-bottom, this unified scheduling resolves discontinuities at critical transition points (e.g., $t = 4$), enabling smooth latent evolution and high-fidelity photorealistic identity reconstruction. The improved alignment between $z_1^{(0)}$ and $\bar{z}_1^{(0)}$ validates the effectiveness of our hybrid-sampler scheduling in mitigating reconstruction artifacts.

## 2.4 Entanglement: Multi-Element (IP) Entanglement via Attentional Gating

As shown in Fig. 6, to address **C3**, we initiate generation from shared latent $\bar{z}^{(T)}$, reconstructing $I_1$, $I_2$, and target $I_3$ under null-text constraints $\emptyset_1^{(t)}$ and $\emptyset_2^{(t)}$. During parallel diffusion, we replace attention maps of $I_3$ with self-attention and cross-attention maps of $I_1/I_2$ for semantic tokens.

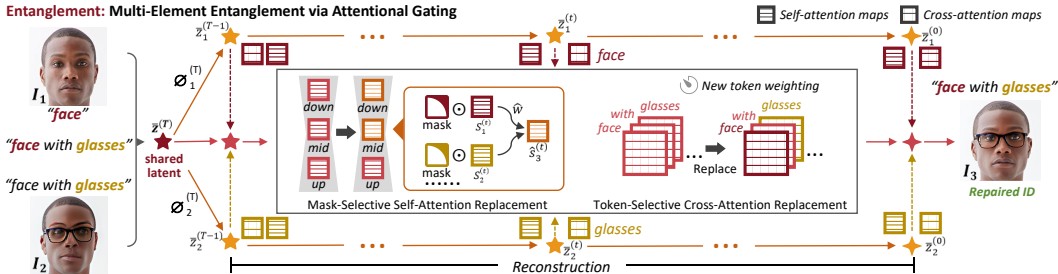

Figure 6: Multi-element entanglement: Mask/Token-selective self/cross-attention replacement.

**Mask-Selective Self-Attention Replacement.** In the step $t$, based on **Observation 3**, self-attention preserves *Single-Element Structures*, we introduce a semantic-guided mask mechanism for self-attention feature selection. Specifically, spatial masks $M_1$ (e.g., "face") and $M_2$ (e.g., "glasses") are applied to self-attention maps $S_1$ (from $I_1$) and $S_2$ (from $I_2$) to isolate target regions:

$$S_3^{(t)} = \sum_{i=1}^{2} S_i^{(t)} \odot W_i + S_3^{(t)} \odot W_3, \tag{7}$$

For non-target regions $W_3$, the original self-attention map $S_3$ of target image $I_3$ is retained. The effective regions $W_1$ and $W_2$ for filtering $S_1$ and $S_2$ are defined as:

$$\begin{aligned} W_1 &= & M_1 \cap (1 - M_2) + \hat{w} \, (M_1 \cap M_2), \\ W_2 &= & M_2 \cap (1 - M_1) + (1 - \hat{w}) \cdot (M_1 \cap M_2), \, W_3 = \quad 1 - (M_1 \cap M_2). \end{aligned} \tag{8}$$

Physical interpretation: Exclusive zones of $M_1/M_2$ retain full self-attention from $S_1/S_2$. Overlapping zones apply weighted fusion with coefficient $\hat{w}$. The entangled $S_3^{(t)}$ undergoes row normalization for valid attention distributions, with replacement confined to down/mid U-Net layers to preserve inter-element generability (Analysis in Appendix A.5).

**Token-Selective Cross-Attention Replacement.** At step $t$, based on **Observation 3**, cross-attention facilitates *Multi-Element Interactions*, we utilize visual attention maps $A^{(t)}$ (Fig. 3(2)) to selectively entangle features between $I_1$ and $I_2$. Let $A_1$, $A_2$, $A_3$ denote cross-attention maps for $I_1$, $I_2$, $I_3$, respectively. The replacement rule can be defined as:

$$A_3^{(t)}[i,j] = \mathbf{1}_{\{i \in \mathcal{T}_1\}} A_1^{(t)}[i] + \mathbf{1}_{\{j \in \mathcal{T}_2\}} A_2^{(t)}[j] + \mathbf{1}_{\{i \notin \mathcal{T}_1, j \notin \mathcal{T}_2\}} A_3^{(t)}[i,j]. \tag{9}$$

where $i, j \in [0, 76]$ (maximum length of text token sequence). Target token maps (e.g., "face" from $I_1$, "glasses" from $I_2$) are replaced throughout $t \in [0, T]$ to ensure semantic coherence. $\mathbf{1}_{\{condition\}}$: Indicator function (1 if true, else 0); $\mathcal{T}_1, \mathcal{T}_2$: Target token sets for source images; $A^{(t)}[i]$: Cross-attention map for the $i$-th token. By integrating BlendDiffusion Avrahami et al. (2022), our framework maintains source-specific structural priors and facilitates token-wise, context-aware interactions, enabling identity-consistent restoration without extra training.

## 3 EXPERIMENT

Experimental setup in Appendix A.6, and hyperparameter sensitivity analysis in Appendix A.8.

**Comparisons.** As shown in Fig. 7 and Tab. 1, we compare EditedID with SOTA methods using samples see Appendix A.6 for details. The comparison spans four categories: Identity-Preserving, Identity Fusion, Blind Restoration, and Face-Swapping. We use three metrics: ID-Sim (ID similarity), CLIP-S (edited element IP preservation), and I-Reward (human-expectation compliance excluding artifacts). Fig. 7 includes challenging scenarios (details in Appendix A.7): focused (single face area $> 10\%$ of image, Lines 1,2,4), multi-angle and complex lighting (Line 3), multi-person ID-specific optimization (Line 4), multi-element IP preservation (Line 5), and non-focused scenes (single face area $< 10\%$ of image, Line 6). Key observations are summarized as follows:

❶ *EditedID resolves the trade-off between Original ID and edited Element IP preservation.* Due to *Cross-source Feature Contamination* between the Original ID and Element IP, existing methods

Table 1: **Comparison with ID Preservation Methods**: ID Similarity ( Deng et al. (2019)), IP Preservation (CLIP-S Hessel et al. (2021)), Human Preference (I-Reward Xu et al. (2023)).

| ID Preservation Method | ID-Sim↑ | CLIP-S↑ | I-Reward↑ |
|---|---|---|---|
| Rosberg et al. (2023) | 0.36 | 25.71 | 1.45 |
| Ye et al. (2023a) | 0.35 | 20.42 | 1.02 |
| Zhao et al. (2023) | 0.49 | 25.85 | 1.53 |
| Han et al. (2024) | 0.40 | 26.13 | 1.61 |
| Lin et al. (2024) | 0.34 | 25.43 | 1.65 |
| Wang et al. (2025) | 0.63 | 26.11 | 1.56 |
| Baliah et al. (2025) | 0.41 | 27.63 | 1.68 |
| DeepFaceSwap AI (2025) | 0.52 | 28.02 | 1.69 |
| Ye et al. (2025) | 0.65 | 26.11 | 1.73 |
| **EditedID (Ours)** | **0.73**$_{\uparrow 0.27}$ | **28.14**$_{\uparrow 2.43}$ | **1.82**$_{\uparrow 0.27}$ |

Table 2: **Comparison with Multimodal Editing Large Models.** In-Con denotes In-ContextEdit.

| Multimodal Model | ID-Sim↑ |
|---|---|
| InstructPix2Pix | 0.37 |
| In-ContextEdit | 0.56 |
| GPT-4o Plus | 0.58 |
| Flux.1 Kontext | 0.55 |
| Doubao | 0.63 |
| Dreamina AI | 0.64 |
| BAGEL | 0.50 |
| Qwen-Image-Edit | 0.52 |
| **In-Con w/ EditedID** | **0.72**$_{\uparrow 0.16}$ |
| **Doubao w/ EditedID** | **0.75**$_{\uparrow 0.12}$ |

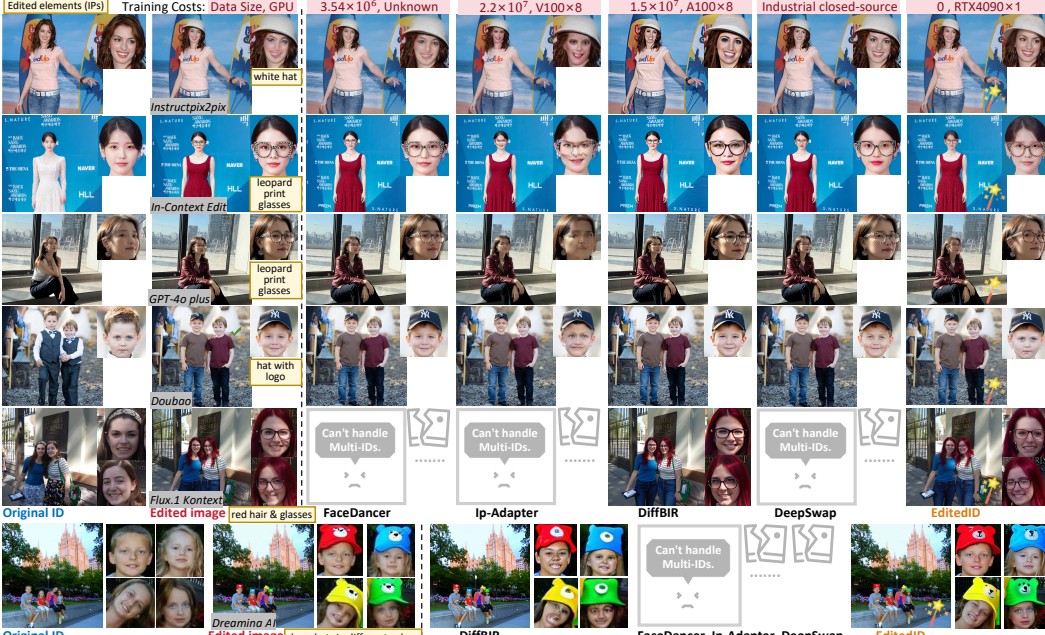

Figure 7: **Qualitative evaluation** of EditedID *vs.* Industrial/Academic facial reconstruction models with training cost comparison (data size, GPU). Edited image from six multimodal editing large models. Yellow boxes indicate edited elements (IP) that need to be preserved in the repaired ID.

exhibit limitations: IP-Adapter often leads to *facial collapse*, DiffBIR corrupts IP (e.g., transform *leopard-print frames* into plain black), and FaceDancer fails to preserve *high-dimensional identity features*. By isolating *Cross-source Feature Contamination* through Hybrid Solver and Attentional Gating, EditedID achieves robust and controllable identity migration while faithfully preserving all edited IP attributes, e.g., *patterns*, *logos*, *colors*. Quantitative evaluations demonstrate consistent improvements over SOTA methods, with average gains of **0.27** in **ID-Sim** and **2.43** in **CLIP-S**.

❷ *EditedID demonstrates robustness in Real-world scenarios.* Existing methods often suffer from performance degradation in real-world scenarios (e.g., profile views, challenging lighting, multi-person scenes, etc.), due to being trained or fine-tuned on facial datasets dominated by professional frontal portraits. The resulting *Cross-source Distribution Bias* restricts their learnable knowledge and generalizability under real-world scenarios: IP-Adapter fails on 45° profile views, while DeepSwap loses edited IP elements and exhibits identity drift under occlusion. In contrast, EditedID mitigates *Cross-source Distribution Bias* via Adaptive Mixing, achieving stable, artifact-free reconstruction in practical scenarios—yielding an average **I-Reward** improvement of **0.27** over baselines.

❸ *Existing methods struggle with Multi-IDs restoration.* Due to the inherent limitations in the application scenarios assumed by existing methods (FaceDancer, IP-Adapter, DeepSwap), they struggle to perform parallel multi-person ID restoration. In contrast, EditedID enables concurrent ID-consistent facial recovery across multiple scenarios—including multi-person focused, non-focused, ID-specific optimization, and multi-element IP preservation—through parallel identity restoration on facial patches. It also enhances original ID resolution in low-resolution defocused scenarios.

In Fig. 7-top, EditedID requires neither complex fine-tuning/training nor labor-intensive data collection, achieving training-free implementation with a single GPU. Additionally, to further evaluate the ID consistency enhancement by EditedID on multimodal editing large models, we compare it against academic Brooks et al. (2023); Zhang et al. (2025) and industrial state-of-the-art models Black Forest Labs (2025); OpenAI (2025); Deng et al. (2025); Wu et al. (2025); ByteDance (2025a;b) in Tab. 2. Clearly, existing large models exhibit facial identity degradation (ID-Sim < 0.7) under various challenging scenarios. This is primarily due to interference from excessive text tokens during long-instruction multi-subject editing, which dilutes the model's focus on visual features and disrupts the retention of original identity priors, often resulting in randomly generated IDs. Notably, EditedID offers flexible compatibility with different large models through external optimization, avoiding model-specific fine-tuning and its associated data/resource costs. After optimization, the academic model In-ContextEdit achieves a **0.16** gain in ID-Sim, while the industrial model Doubao improves by **0.12**. These results demonstrate both the effectiveness and compatibility of EditedID in enhancing ID consistency for multimodal editing large models.

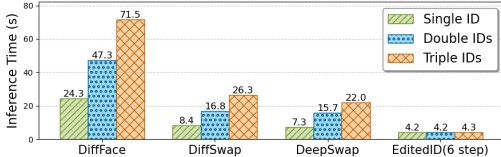
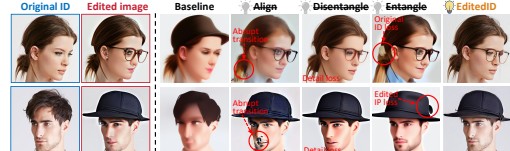

Figure 8: Per Image Time: Single *vs.* Multi IDs.   Figure 9: Qualitative Ablation Evaluation.

**Efficiency.** We evaluate the inference efficiency of EditedID in both single- and multi-ID scenarios (see Fig. 8). For the single-ID case, EditedID achieves an average reconstruction time of approximately 4.2 seconds, which is about 6× faster than the diffusion-based DiffFace (details in Appendix A.11). While baselines exhibit exponential time growth in multi-IDs scenarios, EditedID maintains constant inference time with parallel architecture regardless of ID count.

**Ablations.** EditedID is evaluated through ablation (Fig. 9 and Appendix A.9) on a baseline of null-text inverted real images with prompt-to-prompt identity fusion. **Without alignment** introduces identity mismatches, artifacts, and abrupt transitions, highlighting the role of Adaptive Mixing in mitigating *Cross-source Distribution Bias* during diffusion. **Without disentanglement** produces artifacts or facial distortions, validating the Hybrid Solver's importance in balancing sampling strategies to isolate *Cross-source Feature Contamination* while preserving identity and details. **Without entanglement** causes loss of edited attributes (Elements IP e.g., hats/glasses), underscoring the necessity of Attentional Gating to maintain single-element structures and control multi-element interactions during diffusion. In summary, EditedID is not a simple aggregation of modules, but a coherent diffusion optimization framework. Its effectiveness stems from principled improvements—Adaptive Mixing, Hybrid Solver, Attentional Gating—that optimize the pretrained diffusion process without introducing unnecessary complexity. Ablation studies confirm that each technique provides an essential refinement to diffusion dynamics, enabling stable and interpretable identity reconstruction. Essentially, EditedID is a stable, general diffusion manipulation mechanism.

## 4 CONCLUSION

By reutilizing diffusion dynamics for cross-source feature fusion, we propose EditedID—a training-free and resource-efficient framework for ID-consistent face restoration. We reveal the **limitations of multimodal large models** (requiring extensive data/computation/human efforts) in preserving ID consistency; **plugging in EditedID** significantly enhances ID-consistent capability of multimodal editing large models, demonstrating the **real-world applicability and effectiveness** of EditedID in addressing the ID consistency challenge. Furthermore, with its high ID consistency, EditedID

serves as a pre-edit/post-edit facial **dataset calibration method** to significantly expand existing facial datasets—a single facial sample can yield multiple edited versions, thereby alleviating a **long-standing limitation** in facial ID consistency for multimodal large models—data scarcity and confidentiality. Overall, by revealing **new insights** into diffusion Trajectories, Samplers, and Attention mechanisms, we believe EditedID will underpin future **broader applications and academic research** in multimodal fusion, deformation-resistant editing, and fine-grained attribute preservation.

## ETHICS STATEMENT

This work studies identity-consistent facial restoration and editing techniques. While the proposed method is designed for benign applications such as image restoration, creative content generation, and research purposes, we acknowledge that such technologies could be misused (e.g., for impersonation or privacy invasion). To mitigate such risks, our work does not provide tools for identity acquisition or unauthorized identity manipulation. The method operates only on user-provided images and does not incorporate any identity recognition or retrieval components. All experiments are conducted on publicly available datasets or data with appropriate usage permissions. We emphasize that this research is intended to advance controllable and responsible image generation. We encourage future users and developers to comply with applicable laws and ethical guidelines, obtain proper consent when handling personal data, and avoid malicious applications.

## ACKNOWLEDGMENTS

This work is supported by the National Natural Science Foundation of China under Grants (T2541022, 62361166629), and the Key Research and Development Project of Hubei Province (2022BAD175, 2022BCA009). It is also partially supported by the WHU–Kingsoft Joint Lab. Additional support was provided by the National Natural Science Foundation of China (Grant No. 62571380).

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
