# A APPENDIX

## A.1 RELATED WORK

**Identity-Preserving** approaches Yue & Loy (2022); Ohayon et al. (2024); Ye et al. (2023a; 2025) typically rely on local fine-tuning of diffusion models using edited image pairs. IP-Adapter Ye et al. (2023a) has a significant disparity in training data volume. This leads to the loss of fine-grained identity features such as skin details and hair texture, rendering it ineffective for realistic facial restoration. DreamID Ye et al. (2025) adopts a triplet-based identity supervision strategy within a dual U-Net architecture. While this enables strong stylization capabilities, it suffers from high training complexity and constrained generalization due to limited and structured training data.

**Identity Fusion** approaches Yao et al. (2025); Nam et al. (2024); Tang et al. (2023) primarily target identity transfer in general scenarios. Although FreeGraftor Yao et al. (2025) and DreamMatcher Nam et al. (2024) emphasize editable transfer rather than pure identity transfer, prioritizing flexibility in editing over strict identity fidelity. As a result, these methods perform adequately in perceptually tolerant domains (e.g., animal or cartoon identities), but fall short in perceptually sensitive contexts such as realistic human faces.

**Blind Restoration** methods Yue & Loy (2022); Ohayon et al. (2024); Lin et al. (2024) focus on enhancing fine details in degraded facial images. Early approaches such as Yue & Loy (2022) attempt to bridge the low-to-high quality gap by modeling intermediate diffusion states. However, these methods emphasize restoration over generation and thus struggle to recover severely degraded faces. More recent techniques Ohayon et al. (2024); Lin et al. (2024) are conditioned solely on the degraded target image, they lack explicit guidance for identity preservation. This limitation similarly affects recent diffusion-based super-resolution frameworks Hu et al. (2025); Yang et al. (2024b); Guo et al. (2024); Yang et al. (2024a); Ito et al. (2025), which despite improved perceptual quality and remain deficient in identity-specific guidance.

**Face-Swapping** methods Ye et al. (2025); Wang et al. (2025); Wang (2024); DeepFaceSwap AI (2025) enable arbitrary face replacement in open-world scenarios. However, as the core mechanism, adapting the source identity to the edited target often results in identity feature degradation due to expression changes and facial distortions, thereby yielding low identity consistency. While recent advances such as DreamID Ye et al. (2025) achieve higher identity preservation in a wide range of poses and light conditions, they remain sensitive to input quality and fail to maintain identity fidelity when target faces are heavily degraded or distorted.

## A.2 DIFFUSION SAMPLERS: DDPM, DDIM, DPM-SOLVER++

**DDPM** employs Markov chains. It requires all $T$ steps (typically $T = 1000$), with each step necessitating a call to the noise prediction model $\varepsilon_\theta$, resulting in high computational cost.

**DDIM** employs non-Markovian sampling. It accelerates the generation by skipping intermediate steps (e.g., using only $T = 50$ steps), significantly improving efficiency. Its key mechanisms are:

1) The $\tilde{z}^{(0)}$–$z^{(0)}$ Alignment: Predicts the original latent image $\tilde{z}^{(0)}$ from the noisy latent $z^{(t)}$ at step $t$:

$$\tilde{z}^{(0)} = \frac{z^{(t)} - \sqrt{1 - \bar{\alpha}_t} \cdot \varepsilon_\theta \left( z^{(t)}, t \right)}{\sqrt{\bar{\alpha}_t}}, \tag{10}$$

where $\bar{\alpha}_t$ is the noise schedule parameter. During inversion, this prediction $\tilde{z}^{(0)}$ is forced to align with the real image $z^{(0)}$.

2) The 1st-Order Deterministic Path: Computes the denoised sample $z^{(k)}$ in a previous target step $k$ ($k < t$) deterministically using $z^{(t)}$ and the aligned $\tilde{z}^{(0)}$:

$$z^{(k)} = \sqrt{\bar{\alpha}_k} \cdot \tilde{z}^{(0)} + \sqrt{1 - \bar{\alpha}_k} \cdot \frac{\left( z^{(t)} - \sqrt{\bar{\alpha}_t} \cdot \tilde{z}^{(0)} \right)}{\sqrt{1 - \bar{\alpha}_t}}. \tag{11}$$

where $\bar{\alpha}_k$ is the noise schedule in step $k$. This formula, derived from the first-order Taylor expansion (Euler method), provides a low-order approximation to solve the underlying probability-flow ODE.

**DPM-Solver++** models the reverse process as an Ordinary Differential Equation (ODE), combined with high-order numerical methods to accelerate the solving process and reduce required sampling steps (e.g., $T = 10$ steps). The key processes include:

1) High-Order Taylor Expansion: Uses the high-order Taylor expansion function $Taylor\_Expan(\cdot)$ to approximate $z^{(k)}$:

$$z^{(k)} \approx Taylor\_Expan\left(z^{(t)}, \varepsilon_\theta\left(z^{(t)}, t\right), z_\theta\left(z^{(t)}, t\right)...\right). \tag{12}$$

Approximating the ODE solution with high-order terms enables traversing from $z^{(T)}$ to $z^{(0)}$ in a very few steps with fast sampling.

2) Hybrid Prediction ($\varepsilon_\theta/z_\theta$): $\varepsilon_\theta\left(z^{(t)}, t\right)$ is the noise prediction model, $z_\theta\left(z^{(t)}, t\right)$ is equivalent to Eq. 10 (i.e., $z_\theta \equiv \tilde{z}^{(0)}$). The predicted $\tilde{z}^{(0)}$ directly participates in the denoising process.

### A.3 Analysis of DDIM and DPM-Solver++ Characteristics

DDIM Inversion-Reconstruction preserves ID features but loses details/texture. This is because:

- **Deterministic Path:** During inversion, DDIM forces the predicted ID $\tilde{z}^{(0)}$ to approximate the original $z^{(0)}$ (Eq. 10), thereby "anchoring" the diffusion path to the original ID image $\tilde{z}^{(0)}$ to achieve strong feature preservation.
- **1st-Order Smoothing:** DDIM's 1st-order computation (Eq. 11) uses only the current time step's gradient for a linear approximation. This approach fails to accurately capture rapid, nonlinear changes in pixel values (i.e., high-frequency details) in the high-dimensional space, resulting in smoothed details.
- **Error Accumulation:** The solution of the current latent vector $z^{(k)}$ depends directly on the previous latent $z^{(t)}$ (Eq. 11), leading to gradual error accumulation that further degrades texture quality.

DPM-Solver++ Inversion-Reconstruction generates realistic details/texture in fewer steps but fails to preserve identity, often producing random IDs. This is because:

- **High-Order Taylor Expansion:** DPM-Solver++ (Eq. 12) uses high-order Taylor expansion to compute $z^{(k)}$, which incorporates the rate of change (first-order derivative), the change of that rate (second-order derivative), and higher-order information. This enables a finer and more nonlinear approximation of the data distribution and denoising process, accurately reconstructing rapid pixel variations and facilitating realistic detail and texture generation.
- **Path Deviation:** High-order Taylor expansion acts as an extrapolation technique—predicting future values based on current and historical information. During inversion, this causes the computed path to deviate from the exact $z^{(0)}$ reconstruction path, shifting toward what the model deems "plausible", thereby losing original ID features.
- **Prior Interference:** The hybrid prediction ($\varepsilon_\theta/z_\theta$) means that the $\tilde{z}^{(0)}$ predicted by $z_\theta$ incorporates the model's generative prior rather than strictly adhering to the input image $z^{(0)}$. Using this $\tilde{z}^{(0)}$ during denoising accumulates bias and leads to loss of the original ID.

### A.4 Timesteps Initialization (DDIM *vs.* DPM-Solver++)

**DDIM Timesteps Initialization.** The temporal discretization strategy critically governs the sampling efficiency in diffusion models. DDIM (Denoising Diffusion Implicit Models) employs a linear schedule where time steps $\tau_i$ follow a deterministic uniform decay from $\tau_{max}$ to $\tau_{min}$ according to the specification in Song et al. (2020):

$$\tau_i = \tau_{max} - \frac{i}{T-1}(\tau_{max} - \tau_{min}), \ i \in \{0, 1, ..., T-1\} \tag{13}$$

with boundary parameters $\tau_{\max} = 0.9999$ and $\tau_{\min} = 0.0001$. This generates a strictly linear sequence where $\triangle\tau = \tau_i - \tau_{i+1}$ remains constant throughout all steps, producing a uniform coverage of the diffusion trajectory.

**DPM-Solver++ Timesteps Initialization.** DPM-Solver++ fundamentally diverges through its log-uniform schedule Lu et al. (2022), defined by exponential decay of time steps $\sigma_i$:

$$\sigma_i = exp(\ln \sigma_{max} + \frac{i}{T-1}(\ln \sigma_{min} - \ln \sigma_{max})), \tag{14}$$

parameterized with $\sigma_{\max} = 0.99$ and $\sigma_{\min} = 0.01$. This configuration yields geometrically shrinking intervals $\triangle \sigma = \sigma_i - \sigma_{i+1}$ that concentrate step density near $t \to 0$, aligned with regions of highest curvature in the reverse diffusion ODE. The distinct homogeneity profiles arise from the core design philosophies: DDIM's linear spacing ensures predictable convergence for moderate-step sampling, while DPM-Solver++'s exponential compression maximizes information capture during critical early denoising phases for few-step efficiency. Both methods preserve boundary consistency with $\tau_0 = \sigma_0 \approx 1$ (pure noise) and $\tau_{T-1} = \sigma_{T-1} \approx 0$(clean data), yet their contrasting time warping—uniform versus logarithmic—reflects divergent approaches to temporal resolution allocation.

## A.5 SELF-ATTENTION IMPACTS IN DIFFERENT LAYERS

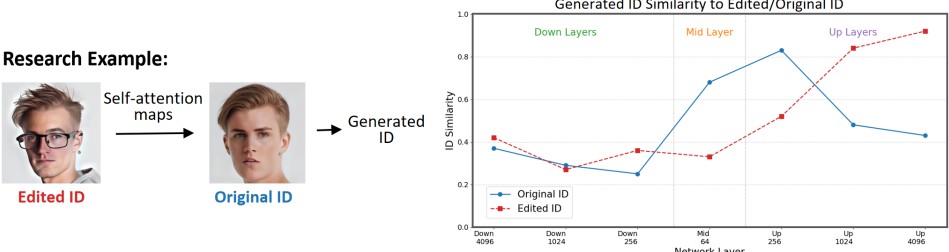

Figure 10: Layer-Specific Attention Replacement(Dual-Image Fusion).

The U-Net architecture used for noise prediction is illustrated in Fig. 3(1). Our primary focus lies on the attention layers, which comprise the downsampling blocks, the middle connection block, and the upsampling blocks. Each block contains cross-attention and self-attention maps of varying resolutions. The feature resolutions of the downsampling and upsampling blocks are 256, 1024, and 4096, while the middle block operates at a resolution of 64.

To investigate the impact on reconstruction fidelity when replacing self-attention maps from different layers within this U-Net structure, as illustrated in Fig. 10, we substitute the self-attention maps in the Original ID with corresponding maps sourced from the Edited ID during the reconstruction of the Original ID. We employ ArcFace Deng et al. (2019) to extract identity similarity by computing the cosine distance between the Generated ID (produced after sequentially replacing each layer) and both the Original ID and the Edited ID. Crucially, within a single generation process, only the self-attention maps belonging to a specific layer resolution are replaced, and this replacement is applied consistently throughout all diffusion steps of the generation.

The investigation of self-attention reveals two key findings: 1) Larger size attention maps capture higher-level semantics and exert greater influence on the output. 2) Replacing attention in the upsampling blocks has a more pronounced effect compared to replacements in the downsampling or middle blocks. Upsampling block replacement enhances fine-grained feature preservation but constrains generation flexibility, whereas downsampling block replacement increases generation flexibility at the cost of potential feature degradation and reduced fidelity. These insights informed our design of the disentanglement module. To enable Mask-Selective Self-Attention Replacement to mediate interactions between source objects, based on our experimental observations, self-attention replacement is exclusively applied in the downsampling and middle blocks. Replacement in the 256-resolution layer of the upsampling block is only sporadically incorporated.

## A.6 EXPERIMENTAL SETUP

**Test Datasets.** Since our method does not require a training phase, large-scale training datasets are unnecessary. To evaluate practical utility in challenging real-world scenarios, our test datasets comprise: Two open-source datasets: CrowdHuman (Shao et al., 2018) and DeepFashion-MultiModal

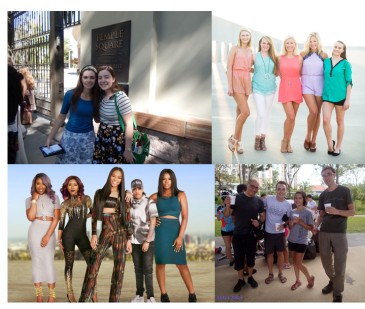 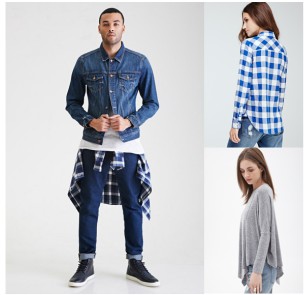 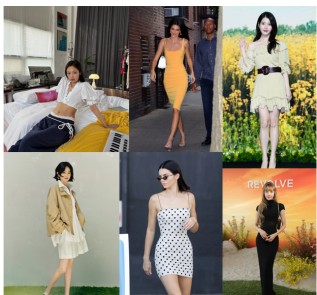

**CrowdHuman** (Shao et al. 2018)   **DeepFashion-MultiModal** (Jiang et al. 2022)   **Open-World Social Media (Xiaohongshu/YouTube)**

Figure 11: Composition of the test dataset.

(Jiang et al., 2022); Open-World Social Media (Xiaohongshu/YouTube) samples. As illustrated in Fig. 11. CrowdHuman provides crowd-sourced images (primarily from Google Search) featuring multiple individuals with various occlusions. DeepFashion-MultiModal contains single-person fashion images with varied facial angles, enabling robustness evaluation against pose variations. Open-World Social Media covers celebrity images (single/multi-person) with complex environmental interference. All 1,200 test samples are carefully curated to span challenging dimensions: multi-angle faces, complex lighting, facial occlusions, and variable ID counts. Distribution: 30% CrowdHuman, 35% DeepFashion-MultiModal, 35% Open-World Social Media.

**Baselines.** Our ablation baseline integrates Prompt-to-Prompt Hertz et al. (2022) and Null-Text Inversion Mokady et al. (2023), reflecting the foundations of our method. First, Null-Text optimization derives latent starting points for both the original identity (ID) and the edited image. Reconstruction then proceeds from these starting points. For the target image, diffusion starts from the latent starting point of the original ID. Following Prompt-to-Prompt, cross-attention maps corresponding to target elements from both the original ID and edited image reconstructions replace the target image's attention maps during diffusion. This iterative process yields the target image with cross-source attention guidance. For a comprehensive evaluation, we compare four state-of-the-art categories: Identity-Preserving Methods: Ip-Adapter, DreamID Ye et al. (2025); Identity Fusion Methods: FaceDancer Rosberg et al. (2023), FaceAdapter Ye et al. (2023a); Blind Restoration Methods: DiffBIR Lin et al. (2024); Face-Swapping Methods: DeepSwap DeepFaceSwap AI (2025), DiffSwap Zhao et al. (2023), CSCS Wang et al. (2025), REFace Baliah et al. (2025).

**Implementation Details.** Unlike existing methods requiring high-spec training/deployment hardware (e.g., 8×A100/V100), our approach eliminates training and operates on a single NVIDIA RTX 3090/4090. All experiments are performed on a single NVIDIA RTX 4090 (24GB VRAM). We use Stable Diffusion v1.4[1] with target object masks generated by the FaRL Zheng et al. (2022) face segmentation method (supporting user customization). The key parameters are configured as follows: total diffusion steps = 6, initial adaptive mixing weight $\lambda_0 = 0.04$, DPM-Solver++ range $[s_1, s_2] = [1, 3]$. The weight for the overlapping region fusion $\hat{w}$ remains user-configurable per scenario to ensure operational flexibility.

**Evaluation Metrics.** ID Similarity (ID-Sim) measures identity preservation between the restored image $I_3$ and the original identity image $I_1$ using ArcFace embeddings Deng et al. (2019):

$$ID\text{-}Sim = cos\left(\mathrm{f}_{ArcFace}\left(I_1\right), \mathrm{f}_{ArcFace}\left(I_3\right)\right), \tag{15}$$

where $\mathrm{f}_{ArcFace}\left(\cdot\right)$ denotes the ArcFace feature extractor and $cos\left(\cdot\right)$ computes cosine similarity. Quantifies whether the restored image retains the original subject's identity. A threshold of $ID\text{-}Sim \geq 0.7$ indicates a successful retention of identity (the same person). The values below $0.7$ signify the identity drift.

CLIP Semantic Similarity (CLIP-S) evaluates preservation of edited attributes (e.g., glasses, accessories) in $I_3$. Physical Meaning: Assesses fidelity to user-specified edits. Higher values ($\sim 20$)

---

[1]https://huggingface.co/CompVis/stable-diffusion-v1-4

indicate better retention of edited elements Hessel et al. (2021). Image Reward (I-Reward) indicates the human perceptual preference for $I_3$ using a learned model Xu et al. (2023).

$$I\text{-}Reward = \mathcal{R}_\theta \left( I_3 \right), \tag{16}$$

where $\mathcal{R}_\theta$ is a ResNet-50 He et al. (2016) based preference predictor trained on $\sim 137$k human judgments. Physical Meaning: Estimates visual naturalness and freedom from artifacts (e.g., distortions, blurring). Higher scores indicate better alignment with human aesthetic standards Xu et al. (2023).

**General Editing Model.** To authentically demonstrate the identity (ID) consistency preservation challenges faced by state-of-the-art image editing systems, our edited images originate from four leading industrial closed-source models: GPT-4o Plus OpenAI (2025), Doubao ByteDance (2025a), Flux.1 Kontext Black Forest Labs (2025), and Dreamina AI ByteDance (2025b); alongside open-sourced academic models: InstructPix2Pix Brooks et al. (2023) and In-Context Edit Zhang et al. (2025). Through qualitative evaluation, we reveal significant ID consistency limitations in the outputs from these heavily-resourced models (trained with massive datasets/computational resources). We further demonstrate that integrating our training-free EditedID framework substantially enhances ID preservation capabilities across these editing platforms, thereby validating both the applicability of our approach and its effectiveness in resolving this core challenge.

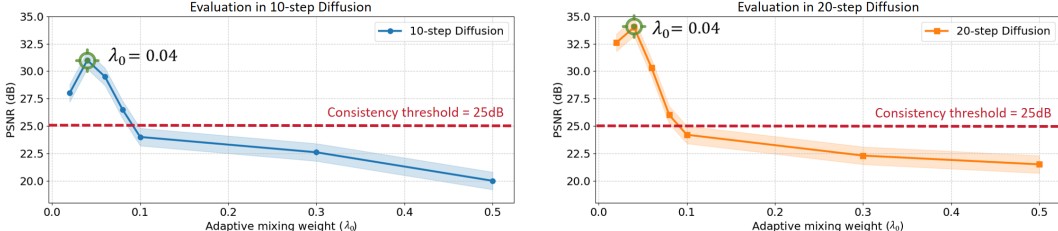

Figure 12: Quantitative evaluation of reconstruction consistency (PSNR) under different adaptive mixing weight $\lambda_0$ settings. The consistency threshold between reconstructed and original images is 25dB (higher PSNR indicates better quality). Results demonstrate that $\lambda_0 = 0.04$ achieves optimal performance across different diffusion steps.

## A.7 FACIAL CORRUPTION AND CONSISTENCY CRITERIA ACROSS SCENARIOS

Based on restoration difficulty, we categorize facial corruption in images edited by multimodal large models into two levels: High facial corruption: distorted, missing, or blended facial features making the face unrecognizable; Low facial corruption: facial structures remain intact but exhibit inconsistent or arbitrary identity.

Our test set covers the following challenging restoration scenarios:

- Multi-angle facial restoration: Repair of non-frontal faces (e.g., 45° or 90° profiles, overhead shots) is challenging due to the predominance of professional frontal portraits in training data.
- Complex lighting: Requires maintaining natural lighting and shadow coherence under highly variable illumination.
- Occluded faces: Restoration is performed around obstructions while preserving the occluding objects.
- Multi-person ID-specific optimization: Targets identity corruption which may affect only certain subjects, requiring subject-aware restoration.
- Multi-person multi-attribute IP preservation: Retains multiple personalized element attributes (e.g., burgundy hair and black-frame glasses) during identity recovery.
- Focused scenes (single face area $> 10\%$ of image): Clear facial regions facilitate editing using multimodal large models, typically resulting in low corruption, though may exhibit local inconsistencies or random IDs.
- Non-focused scenes (single face area $< 10\%$ of image): Low facial visibility editing using multimodal large models often leads to high corruption, including distortion or loss of features.

## A.8 SENSITIVITY

The key hyperparameters are assigned to the values as follows: the initial value of the adaptive mixing weight $\lambda_0$ ($t = 0$) of the alignment module, the DPM-Solver++ range $[s_1, s_2]$ from the disentanglement module, and the overlapping region fusion weight $\hat{w}$ from the entanglement module. The optimal hyperparameters for null-text optimization are set to Mokady et al. (2023). Since the face-inpainting method requires strict consistency between the target element and the original element, the cross-attention and self-attention maps corresponding to the target element token are replaced throughout all diffusion steps to achieve a strictly consistent output.

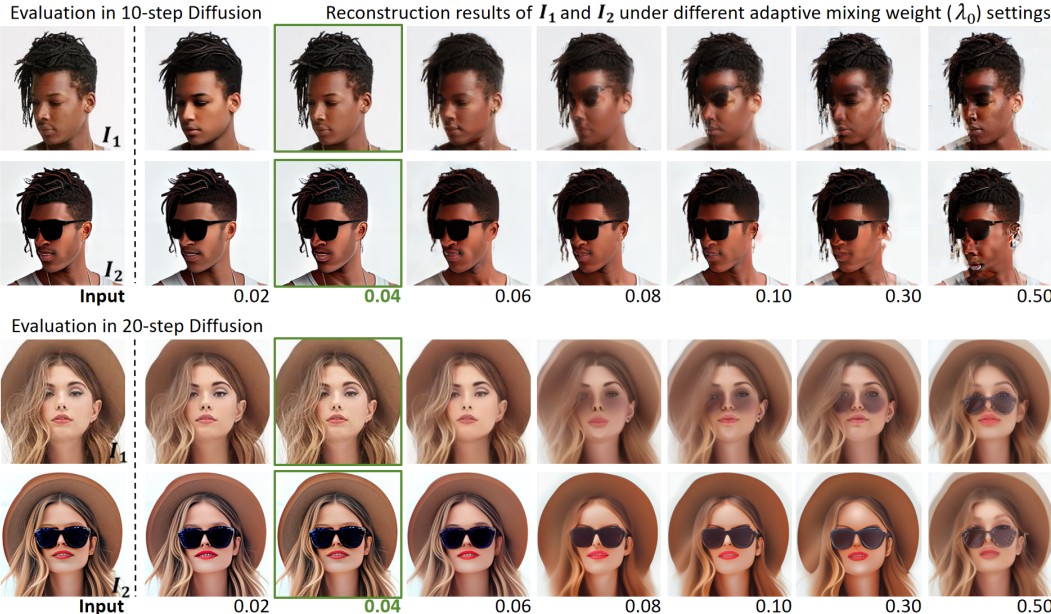

Figure 13: Qualitative evaluation of reconstruction consistency (PSNR) under different adaptive mixing weight $\lambda_0$ settings. Optimal performance is achieved at $\lambda_0 = 0.04$.

**Hyperparameter 1: adaptive mixing weight** $\lambda_0$ ($t = 0$). The adaptive mixing weight $\lambda_0$ ($\in [0, 0.5]$), sourced from the alignment module, governs the latent space alignment between the original identity ($I_1$) and the edited image ($I_2$). This alignment is achieved through the gradient descent from the initial value of $\lambda_0$ at each diffusion step convergent to 0.5, where Eq. 2 confirms $I_1$-$I_2$ alignment. The initial $\lambda_0$ critically impacts the magnitude of the update per step: excessively large values cause disruptive updates and feature degradation, while overly small values impede smooth alignment, forcing damaging abrupt transitions near convergence. Consequently, $\lambda_0$'s initialization directly affects the feature consistency between the disentangled/reconstructed images ($I_1'$, $I_2'$) and the inputs ($I_1$, $I_2$).

To identify $\lambda_0$'s optimal initialization in diffusion steps (10-step, 20-step), we quantitatively evaluated the quality of reconstruction under varying initial $\lambda_0$ using PSNR. PSNR (Peak Signal-to-Noise Ratio) measures the consistency between features in source images ($I_1$, $I_2$) and reconstructions ($I_1'$, $I_2'$) from the aligned latent space. We set a PSNR threshold of 25dB, where the value $\geq$ 25dB indicates the effective preservation of fine-grained features. Smaller initial $\lambda_0$ values mitigate excessive per-step latent updates and feature degradation during alignment, favoring gradual descent for better feature retention. Thus, our analysis focuses on the initialization of $\lambda_0$ within $0 - 0.1$. Tested values: 0.02, 0.04, 0.06, 0.08, 0.1, 0.3, 0.5.

Quantitative results for the initial $\lambda_0$ value (Fig. 12) show that within the range of 0 to 0.1, the reconstruction consistency remains acceptable (PSNR $>$ 25 dB) in different diffusion steps. Consistency degrades as the initial value of $\lambda_0$ approaches 0.5. Consequently, qualitative results (Fig. 13) reveal that the larger initial value of $\lambda_0$ hinders the feature disentanglement between input images ($I_1$, $I_2$), leading to uncontrolled fusion with significant artifacts. At an initial value of $\lambda_0$ at 0.05, the features

of $I_1$ and $I_2$ exhibit chaotic entanglement, losing original characteristics and producing random output. This validates that excessively large initial value of $\lambda_0$ causes destructive updates that blend multi-source features during gradient descent, resulting in mixed features in the final reconstruction. Conversely, overly small initial values (e.g., slightly above 0) cause an excessively slow gradient descent. The resulting sharp alignment shift near convergence prevents faithful reconstruction, causing detail loss (e.g., PSNR drop and facial ID detail degradation at $\lambda_0$=0.02, deviating from inputs $I_1$, $I_2$).

Quantitative and qualitative analysis show that 0.04 is the optimal initial value for the adaptive mixing weight $\lambda_0$. This also demonstrates the important role of the proposed adaptive mixing weight $\lambda_t$ within both the alignment and disentanglement modules.

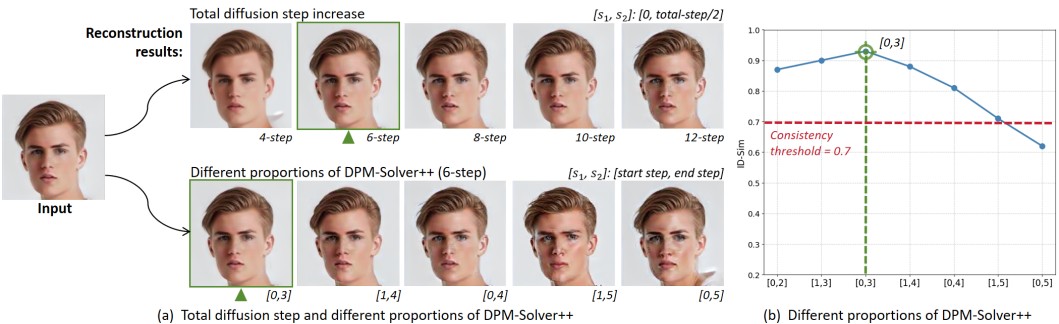

Figure 14: Impact of different diffusion steps and DPM-Solver++ proportions on the reconstruction results. $[s_1, s_2]$: starting and ending diffusion steps for DPM-Solver++ invocation. Results show that consistent input-reconstruction alignment is maintained within just 6 diffusion steps when applying DPM-Solver++ at $[0, 3]$.

**Hyperparameter 2: DPM-Solver++ range** $[s_1, s_2]$. The DPM-Solver++ invocation range $[s_1, s_2]$ ($s_1 < s_2$; $s_1, s_2 \in [0, T)$), sourced from the disentanglement module (Eq. 5), controls the step range and position where the Hybrid Solver invokes DPM-Solver++ during diffusion. Our objective is to optimize the reconstruction path via hybrid DPM-Solver++/DDIM sampling, achieving optimal fidelity to input ID features and facial details. This optimized path is then utilized in the entanglement module.

To investigate the optimal efficiency of achieving the desired reconstruction quality and the best DPM-Solver++ range, qualitative results (Fig. 14(a)-top) demonstrate the reconstructions under different total diffusion steps (4, 6, 8, 10, 12), with $[s_1, s_2]$ uniformly set to $[0, T/2]$. Evidently, reconstructions achieve acceptable fidelity in just 6 steps, exhibiting high ID consistency to the input and minimal artifacts. Consequently, we set the diffusion step count for our method to 6 as the minimum viable steps, ensuring optimal efficiency with an average inference time of 4.2 seconds.

To further analyze the optimal DPM-Solver++ range $[s_1, s_2]$ under the minimum diffusion step setting (6-step), we test various start-end invocation ranges (Selected cases are shown in Fig. 14(a)-bottom): $[0, 3]$, $[1, 4]$, $[0, 4]$, $[1, 5]$, $[0, 5]$. We observe that positioning DPM-Solver++ towards the end of the reconstruction process (closer to $\bar{z}^{(0)}$) yields superior results. This strategy leverages DDIM in the early stages to effectively preserve ID features, while employing DPM-Solver++ later reinforces ID feature details, enabling realistic facial generation in minimal steps. However, allocating an excessively large proportion of steps to DPM-Solver++ (e.g., $[0, 5]$ or $[1, 5]$) can cause ID deviation and introduce artifacts/facial blotches due to over-amplified details. Therefore, we assign DPM-Solver++ with a small number of steps in reconstruction. This optimizes detail generation and accelerates convergence without compromising original ID features. Based on this analysis and quantitative validation in Fig. 14(b), setting $[s_1, s_2]$ to $[0, 3]$ effectively balances DDIM's ability to preserve original identity features and DPM-Solver++'s capacity for detailed refinement, thereby achieving optimal ID consistency.

This qualitative analysis of reconstruction efficiency and the optimal DPM-Solver++ range $[s_1, s_2]$ demonstrates the crucial role of the proposed hybrid solver in improving both reconstruction efficiency and quality.

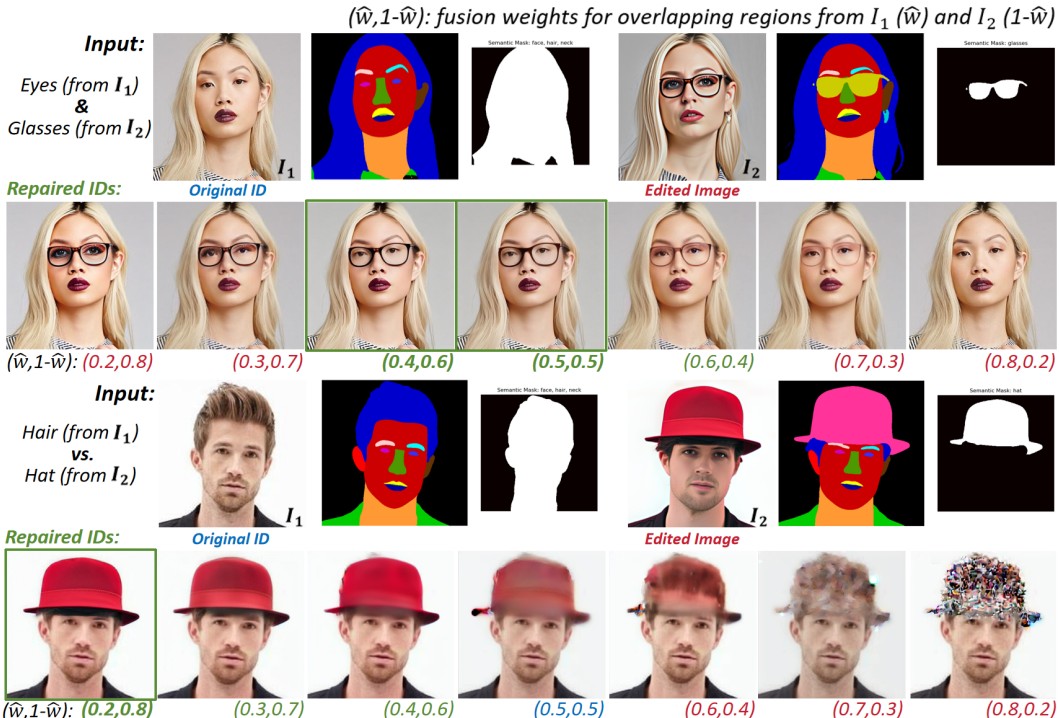

Figure 15: Inpainting results under different fusion weight $\hat{w}$ across various scenarios. $\hat{w}$: fusion weight for overlapping regions of original identity image $I_1$; 1-$\hat{w}$: fusion weight for overlapping regions of edited image $I_2$. Green color indicates physically plausible ranges (e.g., hat above hair), and red color indicates implausible ranges (e.g., hair above hat). Total diffusion steps are 6.

**Hyperparameter 3: overlapping region fusion weight $\hat{w}$.** In the entanglement module, to selectively maximize consistency between facial ID and edited elements (IP), we perform full selective replacement of self-attention for non-overlapping regions within the target object mask (extracted via the FaRL face segmentation method Zheng et al. (2022), supporting user customization), as defined in Eq. 8. For overlapping regions within the target mask (e.g., transparent glasses), we aim to preserve the edited element (e.g., frames) while restoring the original ID (e.g., eyes). Thus, we employ the overlapping region fusion weight $\hat{w}$ ($\hat{w} \in [0, 1]$), where $\hat{w}$ denotes the weight assigned to elements from the original ID ($I_1$) in the overlapping region, and $1 - \hat{w}$ denotes the weight assigned to elements from the edited image ($I_2$).

In particular, the optimal value of $\hat{w}$ varies in different scenarios and it is user-configurable, primarily dictated by its physical interpretation in the target use case. To investigate the setting principles for $\hat{w}$ in different applications (Fig. 15), we visualize the target element masks from $I_1$ and $I_2$ and show the inpainting results under various ($\hat{w}, 1 - \hat{w}$) settings: (0.2, 0.8), (0.3, 0.7), (0.4, 0.6), (0.5, 0.5), (0.6, 0.4), (0.7, 0.3), (0.8, 0.2). We categorize the application scenarios into two types: (1) Co-existence: Elements from different sources need simultaneous preservation (e.g., Eyes ($I_1$) & Glasses ($I_2$)); (2) Layer Coverage: Elements from different sources exhibit an occlusion relationship (e.g., Hair ($I_1$) *vs.* Hat ($I_2$)).

Qualitative evaluation reveals the setting principles for $\hat{w}$: **1) Co-existence Scenarios**, where optimal values approach $\hat{w} \approx 1 - \hat{w}$. Specifically, when preserving eyes ($I_1$) and glasses ($I_2$) concurrently, the green-marked interval from (0.4, 0.6) to (0.6, 0.4) generally contains the best values. Since both elements are equally important, balanced ($\hat{w}, 1 - \hat{w}$) weights best preserve cross-source features. In contrast, extreme values cause excessive glass retention or loss. **2) Layer Coverage Scenarios:** Optimal values satisfy the underlying element weight ($\hat{w}$) < the top element weight ($1 - \hat{w}$). For hair ($I_1$, underlying) and hat ($I_2$, top), the green-marked interval (0.2, 0.8) to (0.4, 0.6) produces plausible results. The setting of $\hat{w} < 1 - \hat{w}$ aligns with reality (hair under the hat). The

red-marked interval from (0.6, 0.4) to (0.8, 0.2) violates diffusion priors (implausible hat-under-hair configuration), generating artifacts and blotches.

This analysis demonstrates that the proposed attentional gating is effective in selective feature fusion. The established $\hat{w}$ configuration principles provide user-actionable guidance, ensuring method flexibility in real-world applications.

## A.9 QUANTITATIVE ABLATION STUDY

Quantitative ablation results in Tab. 3 show that removing the alignment module reduces identity consistency (-0.16 ID-Sim), disabling the disentanglement module degrades perceptual realism (-0.13 I-Reward), and eliminating the entanglement module causes the most significant drop in edited attribute preservation (-2.81 CLIP-S). These demonstrate the effectiveness of the proposed components—adaptive mixing, hybrid solver, and attentional gating—in optimizing diffusion-based facial reconstruction across three key dimensions: identity consistency, realism, and element preservation.

Table 3: Quantitative ablation study results

| Model | ID-Sim↑ | CLIP-S↑ | I-Reward↑ |
|---|---|---|---|
| **Full Model** | **0.73** | **28.14** | **1.82** |
| w/o Alignment | 0.57 | 26.92 | 1.72 |
| w/o Disentanglement | 0.63 | 26.48 | 1.69 |
| w/o Entanglement | 0.61 | 25.33 | 1.78 |
| Baseline | 0.21 | 19.32 | 1.03 |

## A.10 USER SURVEY

To further evaluate the facial reconstruction performance of EditedID, we conducted a user study with 200 participants, including computer vision engineers, graphic designers, and lay users. This stratified sampling ensured a balanced assessment across both usability and technical stability. Participants were asked to rate the restored images on three metrics: ID Consistency, Facial Fidelity, and IP Preservation, on a 5-point scale (higher is better). The average results, shown in Tab. 4, clearly demonstrate that our method achieves competitive outcomes across all three dimensions compared to existing approaches.

Table 4: The participants were asked to rate: (1) ID Consistency, (2) Facial Fidelity, and (3) IP Preservation. The perfect score is 5.

| Method | ID Consistency↑ | Facial Fidelity↑ | IP Preservation↑ |
|---|---|---|---|
| Rosberg et al. (2023) | 3.34±0.31 | 3.83±0.07 | 4.19±0.03 |
| Ye et al. (2023a) | 3.16±0.97 | 2.13±0.94 | 2.11±0.33 |
| Zhao et al. (2023) | 3.77±0.52 | 3.62±0.83 | 3.41±0.43 |
| Han et al. (2024) | 3.61±0.02 | 3.51±0.40 | 3.48±0.45 |
| Lin et al. (2024) | 3.11±0.90 | 4.02±0.75 | 3.21±0.70 |
| Wang et al. (2025) | 4.40±0.33 | 3.55±0.02 | 3.62±0.73 |
| Baliah et al. (2025) | 3.32±0.53 | 4.11±0.33 | 3.81±0.93 |
| DeepFaceSwap AI (2025) | 4.22±0.17 | 4.62±0.93 | 4.01±0.69 |
| Ye et al. (2025) | 4.68±0.10 | 4.78±0.37 | 4.32±0.20 |
| **EditedID (Ours)** | **4.85±0.52** | **4.83±0.36** | **4.72±0.19** |

## A.11 INFERENCE COST ANALYSIS

We report approximate inference cost under a reference implementation. EditedID follows the standard two-stage diffusion pipeline, consisting of inversion (alignment) and reconstruction with disentanglement/entanglement, without introducing additional inference stages. The primary efficiency gain comes from reducing diffusion steps, while memory usage remains comparable to existing inversion-based methods. The reported step numbers correspond to the minimal configuration used by EditedID; we also support more flexible or increased step settings to achieve higher-quality results

Table 5: Approximate inference cost of different methods for reference.

| Method | Diffusion Steps | Total Time (s) | GPU Mem (GB) |
|---|---|---|---|
| Prompt-to-Prompt | 50 | ∼8.9 | ∼13.6 |
| Null-text Inversion | 50 | ∼56.5 | ∼16.2 |
| EditedID (ours) | 6 | ∼7.0 | ∼16.5 |

when desired. All measurements are obtained under a specific experimental setup and optimized research implementation; actual runtime may vary depending on hardware and software environments.

## A.12 ATTENTION VALIDATION

To verify the attention mechanism, we conducted a analysis in which attention maps from Image 2 were injected into the reconstruction process of Image 1. Specifically, we replaced (i) self-attention maps (single-element) and (ii) cross-attention maps (multi-element) across different U-Net layers, resolutions, and architectural variants. We then evaluated the impact of these substitutions by measuring structural consistency using SSIM and semantic alignment using CLIP-S between the reconstructed output and the ground-truth Image 2. This controlled setup allows us to rigorously quantify how attention at each level contributes to identity-related structural and semantic information.

The results above reveal a clear and consistent pattern: (1) Self-attention replacements substantially improve single-element structural preservation, with the strongest effects observed in up-blocks and at higher spatial resolutions. (2) Cross-attention replacements markedly enhance multi-element semantic alignment, and follow the same hierarchical progression across layers and resolutions. This trend further supports the generality of our conclusions and demonstrates that the identified behaviors hold consistently across different architectural scales and variants.

Table 6: Attention mechanism analysis.

| Layer | Resolution | Replace Self-attn (SSIM) | | Replace Cross-attn (CLIP-S) | |
|---|---|---|---|---|---|
| | | SD1.4 | SDXL | SD1.4 | SDXL |
| **-** | - | 0.22 | 0.24 | 15.22 | 15.14 |
| **Down** | 32×32 | 0.62 | 0.67 | 19.70 | 21.32 |
| | 16×16 | 0.58 | 0.65 | 18.51 | 20.09 |
| **Mid** | 8×8 | 0.52 | 0.56 | 16.74 | 19.21 |
| **Up** | 16×16 | 0.63 | 0.72 | 20.37 | 23.11 |
| | 32×32 | 0.71 | 0.76 | 22.98 | 25.31 |

## A.13 SOLVER STABILITY

We further evaluate the stability of our hybrid solver using multiple complementary metrics, including cumulative MSE, maximum transition jump, final latent L2 distance, and PSNR. As reported in Tab. 7, our global hybrid strategy achieves performance comparable to single-solver DDIM in terms of trajectory consistency, while yielding improved reconstruction quality. In particular, the proposed hybrid solver maintains low error accumulation and smooth transitions across solver-switching points, indicating stable inversion–reconstruction trajectories. Overall, these results confirm that the hybrid design preserves stability while benefiting from the complementary strengths of DDIM and DPM-Solver++, leading to consistent and high-fidelity reconstruction.

Table 7: Stability of the hybrid solver.

| Method | Cumulative MSE ($\times 10^{-3}$)↓ | maxJump ($\times 10^{-3}$)↓ | latent L2 ($\times 10^{-2}$)↓ | PSNR (dB)↑ |
|---|---|---|---|---|
| **DDIM-only** | 18.5 ± 3.2 | **1.9 ± 0.6** | **1.7 ± 0.4** | 28.6 ± 0.9 |
| **DPM-only** | 24.2 ± 4.5 | 3.7 ± 0.9 | 2.5 ± 0.6 | 27.2 ± 0.8 |
| **Hybrid-fragmented** | 47.1 ± 5.0 | 8.9 ± 2.2 | 4.3 ± 0.8 | 20.1 ± 1.1 |
| **Hybrid-global (ours)** | **17.8 ± 2.6** | 2.1 ± 0.5 | 1.8 ± 0.3 | **29.0 ± 0.9** |

## A.14 IMPACT OF DIFFBIR PREPROCESSING

We further study the influence of DiffBIR when used as a preprocessing step before EditedID. Tab. 8 reports quantitative results under clean and degraded inputs. Applying DiffBIR alone on degraded images leads to a noticeable drop in identity similarity, while combining DiffBIR with EditedID restores ID-Sim close to the clean-input setting. This suggests that EditedID can effectively compensate for identity distortions introduced by restoration models. Qualitatively, this robustness is mainly attributed to (i) reconstructing the target identity along an independent inversion–reconstruction trajectory, (ii) selectively transferring only edited elements from DiffBIR outputs while discarding distorted identity information, and (iii) leveraging the hybrid solver to refine local texture details. Overall, these results indicate that EditedID does not simply inherit artifacts from preprocessing models, but can mitigate identity degradation and improve visual quality.

Table 8: Impact of DiffBIR preprocessing on identity preservation and image quality.

| Method | ID-Sim | ImageReward |
|---|---|---|
| Clean → EditedID (w/o DiffBIR) | $0.74 \pm 0.03$ | $1.82 \pm 0.05$ |
| Degraded → DiffBIR only | $0.36 \pm 0.07$ | $1.65 \pm 0.08$ |
| Degraded → DiffBIR → EditedID | $0.71 \pm 0.04$ | $1.78 \pm 0.05$ |

## A.15 MULTI-REGION EDITING

We report an ablation study on Attentional Gating under increasingly complex editing settings (single, dual, and three attributes). Results are summarized in Tab. 9. Without gating, performance degrades notably as the number of edited regions increases. Introducing Attentional Gating consistently improves both element preservation (measured by CLIP-S) and identity consistency (ID-Sim), with larger gains observed in more complex multi-attribute scenarios. These results suggest that Attentional Gating helps reduce cross-region interference during fusion, leading to more stable element control and identity retention when multiple edits are applied simultaneously.

Table 9: Effect of attentional gating in multi-region editing.

| Editing Attribute | EditedID Architecture | IP (CLIP-S$\geq$0.75) | ID (ID-Sim$\geq$0.70) |
|---|---|---|---|
| Single | full | 92% | 94% |
| | w/o gating | 78% | 74% |
| Dual | full | 87% | 85% |
| | w/o gating | 69% | 66% |
| Three | full | 84% | 81% |
| | w/o gating | 58% | 62% |

## A.16 MULTI-PERSON PROCESSING

Multi-person editing in EditedID is implemented as an external engineering pipeline rather than an intrinsic model component. For this reason, it is not included in the main framework description and is summarized here for completeness. Specifically, a face detector is first applied to locate all individuals in the input image. User instructions, together with positional correspondences, are used to associate edited elements with their respective identities. EditedID is then independently applied to each detected face in parallel to restore identity consistency. Finally, the processed faces are composited back into the original image according to their spatial locations, and an inpainting model is used to smooth interaction boundaries for natural transitions. We will further update an easy-to-use version of this pipeline in the released codebase to improve practical usability.

## A.17 LIMITATIONS AND DISCUSSION

Our method enables customizable facial restoration in edited portraits by manipulating diffusion trajectories across multiple images, with hard-constrained source identity features ensuring strong facial consistency with the original subject. While EditedID supports parameter-tunable restoration,

its training-free nature introduces a usability barrier, limiting accessibility for non-expert users. Although the method performs reliably across a wide range of open-world fashion portrait edits, certain outlier cases may still lead to inconsistent results due to suboptimal hyperparameter configurations.

To enhance the generalization and user-friendliness of EditedID in practical applications, future work under sufficient computational resources may focus on two directions: integrating the framework into multimodal large models or training it on real-world datasets to improve accessibility. Another promising direction is to adopt EditedID as a data generation tool. Inspired by Instruct-Pix2PixBrooks et al. (2023), which uses generated data from Prompt-to-PromptHertz et al. (2022) to train instruction-following stable diffusion models, EditedID can help alleviate the persistent challenge of multimodal large models in facial ID consistency—a critical barrier in real-world applications due to the scarcity and confidentiality of facial data, as well as the difficulty in acquiring aligned pre- and post-edit pairs. With its high ID consistency, EditedID can serve as a calibration model: real face images can be edited and then corrected using EditedID to form a hybrid dataset (pre-edit: real/synthetic; post-edit: generated). Such mixed data improves real-world applicability. Moreover, the diversity of editing instructions enables extensive sample expansion—a single facial sample can yield multiple edited versions, greatly augmenting dataset size and diversity. The resulting large-scale calibrated dataset can circumvent current limitations on facial data usage and be employed to train identity-preserving models or fine-tune multimodal editing models, thereby enhancing ID retention capabilities. We believe that EditedID holds significant value both as an effective ID-consistent facial reconstruction framework and as a data calibration tool. We believe that, with sufficient computational resources in the future, this method will yield more valuable research findings and deliver unexpected surprises in its applications.