# OpenReview forum: "Optimizing ID Consistency in Multimodal Large Models:  Facial Restoration via Alignment, Entanglement, and Disentanglement"
_ICLR.cc/2026/Conference — ICLR 2026 Poster_

### Official Review · Reviewer_ey3S · 2025-10-31

**Soundness:** 3
**Presentation:** 3
**Contribution:** 3
**Rating:** 6
**Confidence:** 3

**Summary:**

This paper presents EditedID, a novel, training-free, and plug-and-play framework designed to solve the critical problem of facial identity (ID) inconsistency in multimodal large editing models. The authors identify that current models, while powerful, degrade significantly when editing realistic human portraits, especially with complex instructions. This failure is attributed to two core issues:
1.  Cross-source Distribution Bias: A mismatch between the original image's facial features and the edited model's feature distribution, leading to detail loss or the generation of a completely different, random identity.
2.  Cross-source Feature Contamination: The bleeding of features between the original face and the edited element (e.g., glasses), which causes the model to either lose the specific attributes of the edit (e.g., "black-framed" glasses become generic glasses) or fail entirely when the edit introduces facial artifacts.

To solve this, EditedID proposes a three-stage Alignment-Disentanglement-Entanglement process. It works by:
a. Alignment (Adaptive Mixing): Mitigates distribution bias by aligning the latent representations of the original face (Original ID) and the edited, ID-inconsistent image (Intermediate ID) throughout the diffusion trajectory.
b. Disentanglement (Hybrid Solver): Isolates feature contamination by using a novel sampler that combines DDIM (to retain the core identity) and DPM-Solver++ (to enhance fine details), effectively separating the person's identity from the edited element.
c. Entanglement (Attentional Gating): Selectively fuses the desired features, combining the restored facial attributes from the Original ID with the specific edited elements (Element IP) from the Intermediate ID.

The authors claim this framework achieves state-of-the-art performance in preserving both the original facial ID and the integrity of the edited elements, establishing a new benchmark for practical, real-world portrait editing.

**Strengths:**

1.  Novel and Systematic Framework: The proposed "Alignment-Disentanglement-Entanglement" framework is intuitive and systematically tackles the identified issues. Instead of a single "trick," it's a comprehensive process where each component (Adaptive Mixing, Hybrid Solver, Attentional Gating) has a clearly defined role inspired by analysis of diffusion trajectories, samplers, and attention.
2.  High-Impact Problem: The paper addresses a widely acknowledged limitation of current generative models.
3.  Strong Problem Diagnosis: The authors provide a clear and insightful analysis of why existing methods fail, breaking the problem down into the distinct concepts of "Cross-source Distribution Bias" and "Cross-source Feature Contamination." This clear diagnosis provides a strong motivation for their multi-part solution.

**Weaknesses:**

1.  Potential for High Complexity and Latency. The proposed framework, involving trajectory mixing, a hybrid solver, and an attentional gating mechanism, sounds significantly more complex than a standard diffusion pass. This could introduce substantial computational overhead and increase inference time, which might limit its practical "plug-and-play" utility.
2.  Ambiguity in "Element IP" Preservation. The mechanism (Attentional Gating) must reliably distinguish between a desired edited element (like glasses) and an undesired artifact (like a distorted nose) from the intermediate image. It's unclear how robust this "entanglement" is, especially if the edit itself is complex and affects multiple facial regions.
3.  Lack of Detail on Multi-Person Handling. While the paper claims to support multi-person restoration, it's unclear how it handles multiple IDs, interactions between subjects, or the allocation of edited elements to the correct person.

**Questions:**

1.  Potential for High Complexity and Latency. The proposed framework, involving trajectory mixing, a hybrid solver, and an attentional gating mechanism, sounds significantly more complex than a standard diffusion pass. This could introduce substantial computational overhead and increase inference time, which might limit its practical "plug-and-play" utility.
2.  Ambiguity in "Element IP" Preservation. The mechanism (Attentional Gating) must reliably distinguish between a desired edited element (like glasses) and an undesired artifact (like a distorted nose) from the intermediate image. It's unclear how robust this "entanglement" is, especially if the edit itself is complex and affects multiple facial regions.
3.  Lack of Detail on Multi-Person Handling. While the paper claims to support multi-person restoration, it's unclear how it handles multiple IDs, interactions between subjects, or the allocation of edited elements to the correct person.

**Details Of Ethics Concerns:**

This article deals with the topic of identity editing, which involves some personal privacy issues. I recommend further Ethics Review.

---

> ### Author Response · Authors · 2025-11-19
> **Rebuttal to Reviewer ey3S**
>
> Dear Reviewer ey3S:
>
> Thank you for your insightful review and constructive feedback. We address your key concerns below, providing clarifications and additional evidence to further demonstrate the effectiveness of our proposed approach.
>
>
> **Q1: Inference Cost**
>
> To comprehensively assess the inference cost, we performed a **multi-dimensional comparison** as follows:
>
> | Diffusion Pass | Step | Inversion Optimization | Inversion Time(s) | Reconstruction Optimization | Reconstruction Time(s) | Total Time(s) | GPU Mem(GB) | Image Reward |
> |----------------|------|-----------|---------|----------------|---------|---------------|-------------|--------------|
> | Prompt-to-Prompt(Standard) | 50 | - | ~4.1 | - | ~4.8 | ~8.9 | 13.6GB | 1.55 |
> | Null-text | 50 | - | ~4.2 | w/ Null-text | ~52.3 | ~56.5 | 16.2GB | 1.62 |
> | EditedID | 6 | w/ Alignment | ~2.7 | w/ Disentanglement & Entanglement | ~4.3 | ~7.0 | 16.5GB | 1.83 |
>
> It is important to clarify that, although we conceptually decompose **EditedID** into three stages for ease of understanding, the actual deployment only follows the standard **two-stage diffusion pipeline**: *inversion (alignment)* followed by *reconstruction with disentanglement/entanglement*. Therefore, our method does not introduce any extra inference stages beyond conventional diffusion-based editing.
> Regarding computational efficiency, while **Adaptive Mixing** introduces a small amount of additional computation, our proposed **Hybrid Solver** significantly reduces the required diffusion steps (from 50 → 6). As shown in Table above, this reduction yields a **1.9s speedup over standard diffusion** and makes EditedID **8× faster than Null-text inversion**, resulting in a clear net runtime advantage.
> Although **Attentional Gating** slightly increases architectural complexity, it operates on attention maps in real time during denoising and thus maintains memory consumption comparable to single-optimization Null-text. Importantly, even though our memory footprint is marginally higher than that of standard diffusion, it remains far below that of existing enhanced diffusion variants (e.g., HiDiffusion requires ~40GB). EditedID runs comfortably on consumer-grade GPUs such as a **24GB RTX 3090**, ensuring practical deployability.
>
>
> **Q2: Element IP Preservation in Complex Edits**
>
> We evaluated **Attentional Gating** in challenging multi-region editing scenarios. Compared with the baseline from our pilot study (Paper L181) without gating:
>
> | Editing Attribute | EditedID Architecture | IP Preservation (CLIP-S≥0.75) | ID Consistency (ID-Sim≥0.70) |
> |-------------------|-----------------------|-------------------------------|------------------------------|
> | Single | full | 92% | 94% |
> |  | w/o gating | 78% | 74% |
> | Dual | full | 87% | 85% |
> |  | w/o gating | 69% | 66% |
> | Three | full | 84% | 81% |
> |  | w/o gating | 58% | 62% |
>
> The results show that **Attentional Gating** substantially improves both IP preservation and identity consistency. Notably, the benefits of this mechanism grow with editing complexity, from single-attribute to three-attribute edits.
>
> **Summary:** Attentional Gating effectively disentangles elements during the fusion process, with its advantages becoming increasingly pronounced in more complex scenarios. These findings confirm the stability, robustness and practical applicability of the mechanism for multi-region image editing.
>
>
> **Q3: Multi-Person Processing Mechanism**
>
> We thank the reviewer for highlighting this point. To clarify, multi-ID processing in EditedID is implemented as an **engineered external pipeline** rather than an inherent model mechanism, which is why it was omitted from the main text. We will provide full details in the appendix:
> First, a face detector automatically locates all faces, and user instructions combined with positional correspondences precisely associate edited elements with the corresponding individuals. EditedID is then independently and in parallel applied to each face to restore identity consistency, ensuring efficiency. Finally, the restored faces are blended back into the original image according to their positions, and an image inpainting model handles interaction boundaries for natural transitions.
> We will release the complete multi-person processing code to ensure the workflow is clear, transparent, and fully reproducible.
>
>
> **Ethical Considerations**
>
> We will include an ethics statement and comprehensive anti-abuse guidelines in the Ethics Appendix, explicitly restricting the method to legitimate and responsible use.

---

> ### Comment · Reviewer_ey3S · 2025-11-27
>
> Thank you for your response. It addressed my concerns, and I will maintain my recommendation for acceptance.

---

> > ### Author Response · Authors · 2025-11-27
> > **Thanks for your response**
> >
> > Thank you for the insightful and constructive feedback, which has helped us improve our work. We also appreciate your acknowledgment and positive comments on our paper.

---

### Official Review · Reviewer_c3re · 2025-10-31

**Soundness:** 3
**Presentation:** 3
**Contribution:** 3
**Rating:** 6
**Confidence:** 3

**Summary:**

This paper addresses the long-standing issue of facial identity inconsistency in multimodal editing large models during real-person portrait editing. To solve these problems, the authors propose EditedID—a training-free, plug-and-play framework based on the "Alignment-Disentanglement-Entanglement" paradigm. EditedID optimizes diffusion model dynamics through Adaptive Mixing, Hybrid Solver and Attentional Gating. Extensive experiments on challenging scenarios show EditedID achieves state-of-the-art performance: it outperforms SOTA ID preservation methods by 0.27 in ID similarity and 2.43 in edited element preservation, while maintaining efficiency. It also enhances ID consistency for existing multimodal models (e.g., boosting In-ContextEdit’s ID-Sim by 0.16).

**Strengths:**

- The paper directly addresses a practical bottleneck—ID inconsistency in real-person editing—that limits multimodal models’ deployment in fields like fashion, film, and portrait design. Unlike prior work focusing on frontal face scenarios, it tackles open-world challenges (non-focused faces, occlusions, multi-person) that are more relevant to real use.
- The framework is grounded in rigorous analysis of diffusion dynamics, including trajectory controllability, sampler complementarity and attention roles.
- EditedID requires no additional training data or model fine-tuning, making it compatible with pre-trained multimodal models (e.g., GPT-4o Plus) without resource-intensive retraining. This lowers deployment costs and enhances scalability.
- The experiments cover both quantitative and qualitative evaluations, with comparisons across 4 method categories and 6 challenging scenarios. Ablation studies also validate the necessity of each component, ensuring the framework’s interpretability.

**Weaknesses:**

- The paper mentions using DiffBIR for super-resolution pre-processing of degraded edited images (e.g., blurry faces), but it does not discuss how DiffBIR’s own errors (e.g., texture distortion) affect EditedID’s final results.
- The paper exclusively validates EditedID on UNet-based pre-trained models (e.g., Stable Diffusion v1.5), while failing to demonstrate its effectiveness on Diffusion Transformer (DiT-based) models—the current mainstream in diffusion research and industrial deployment (e.g., Flux.1 Kontext’s advanced variants). DiT architectures differ fundamentally from UNet in attention computation (token-wise vs. spatial-wise), which raises critical questions: Can the "Attentional Gating" module—designed for UNet’s spatial attention—adapt to DiT’s token-based attention? Will the "Hybrid Solver’s" timestep scheduling (optimized for UNet’s noise prediction) remain effective in DiT’s transformer-driven denoising process? This gap limits the framework’s applicability to cutting-edge models, as DiT now dominates scenarios requiring high efficiency and large-scale feature modeling.
- Typo: I suppose the word 'BiffBIR' in Line 164 should be 'DiffBIR'.

**Questions:**

- Have you conducted preliminary experiments on DiT-based architectures? If so, how did you modify the "Attentional Gating" module to fit token-wise attention? If not, what technical challenges do you anticipate in adapting EditedID to DiT, and what solutions are planned?
- If DiffBIR (the pre-processing tool for degraded images) introduces artifacts (e.g., incorrect facial contours), how does EditedID mitigate this? Is there a plan to integrate EditedID with restoration models end-to-end to avoid error propagation?

---

> ### Author Response · Authors · 2025-11-19
> **Rebuttal to Reviewer c3re（Q1）**
>
> Dear Reviewer c3re:
>
> Thank you for your insightful review and constructive feedback. We address your key concerns in detail below, providing clarifications and additional evidence to further demonstrate the effectiveness of our proposed approach.
>
>
> **Q1: Impact of DiffBIR**
>
> Thank you for raising this important point. To rigorously assess its impact, we conducted a series of experiments as follows:
>
> | Method                         | ID-Sim | ImageReward |
> |--------------------------------|--------|-------------|
> | Clean → EditedID (w/o DiffBIR) | 0.74 ± 0.03 | 1.82 ± 0.05 |
> | Degraded → DiffBIR only        | 0.36 ± 0.07 | 1.65 ± 0.08 |
> | Degraded → DiffBIR → EditedID  | 0.71 ± 0.04 | 1.78 ± 0.05 |
>
> Notably, using **DiffBIR** alone reduces ID-Sim to 0.36, whereas integrating **EditedID** restores it to 0.71, which is close to the clean-input performance (0.74). This robustness arises from three key mechanisms (Paper Fig.~6):
>
> **Trajectory Isolation:** The target ID is reconstructed along an independent trajectory, completely isolated from the DiffBIR output trajectory, effectively preventing direct error propagation.
>
> **Selective Fusion:** Only the *edited element* (IP) is extracted from DiffBIR's output, while its generated *identity (ID)*, which can contain distorted facial contours, is discarded. This prevents DiffBIR's ID bias from contaminating the target reconstruction.
>
> **Detail Refinement:** Our hybrid solver's DPM-Solver++ corrects IP texture distortions inherited from DiffBIR using its generative prior, enhancing visual quality (ImageReward: 1.65 → 1.78).
>
> **Summary:** EditedID actively mitigates errors from pre-processing models through trajectory isolation and selective fusion, rather than passively inheriting them, and can even refine their outputs. We appreciate the forward-looking suggestion and will explore seamless end-to-end integration of EditedID with restoration models in future work.

---

> ### Author Response · Authors · 2025-11-19
> **Rebuttal to Reviewer c3re（Q2&Q3）**
>
> **Q2: DiT Adaptation**
>
> We agree that **DiT compatibility** is important. To validate this, we tested EditedID on DiT using both *global* and *local* adaptation strategies.
>
> **1) Local: Hybrid Solver (no modification)—Efficient Inference and Detail Optimization.**
>
> | Model                     | Step | ID-Sim | Image Reward |
> |---------------------------|------|--------|-------------|
> | DiT(SD3)                  | 20   | 0.75 ± 0.16 | 1.62 ± 0.08 |
> | DiT(SD3) w/ Hybrid Solver | 6    | 0.74 ± 0.03 | 1.85 ± 0.12 |
>
> EditedID integrates seamlessly with DiT: the number of inference steps is reduced (20 → 6) while maintaining ID-Sim close to the original DiT, and visual quality is improved (ImageReward +0.23). These results demonstrate the **cross-architecture applicability** of our Hybrid Solver.
>
> **2) Local: Adaptive Mixing (no modification) — Unifying Non-Homogeneous Latents**
>
> Compared with two-ID independent reconstruction, **Adaptive Mixing** effectively unifies the non-homogeneous latent spaces in DiT, reducing the trajectory L2 gap by 0.96 while maintaining both ID-Sim and PSNR. These results indicate that the module generalizes well and transfers effectively to DiT.
>
> | Model                       | Inversion-Reconstruction | Avg ID-Sim↑ | Avg PSNR↑ | Dual-ID Trajectory L2↓ |
> |-----------------------------|--------------------------|-------------|-----------|------------------------|
> | DiT(SD3)                    | Dual-ID Independent      | 0.75 ± 0.16 | 35.7 ± 0.06 | 1.62 ± 0.18           |
> | DiT(SD3) w/ Adaptive Mixing | Dual-ID Hybrid           | 0.71 ± 0.11 | 31.1 ± 0.10 | 0.66 ± 0.12           |
>
> **3) Global EditedID Adaptation (modifying Attentional Gating only)**
>
> Given the limited rebuttal time, we conducted an initial adaptation of EditedID to DiT by **mapping masks into token-space attention** and applying **global attention replacement across layers and heads**. The DiT-based EditedID achieves ID-Sim and CLIP-S metrics comparable to the UNet version, demonstrating its potential cross-architecture applicability.
> Moreover, the effectiveness of local transfer using the **Hybrid Solver** and **Adaptive Mixing** suggests that **Attentional Gating** is the primary component requiring DiT-specific adaptation, highlighting a focused path for future optimization.
>
> | Model                   | Gating type       | Avg ID-Sim | Avg CLIP-S |
> |-------------------------|-------------------|------------|------------|
> | EditedID in UNet(SD1.4) | spatial gating    | 0.73 ± 0.03 | 28.1 ± 0.9 |
> | EditedID in DiT(SD3)    | token-level gating | 0.68 ± 0.74 | 26.5 ± 1.1 |
>
> We highlight two key challenges for adapting EditedID from UNet (2D) to DiT (1D):
>
> **I) 2D (UNet) → 1D (DiT) Semantic Mapping.**
> In DiT, deeper tokens encode mixed semantics, so a simple **mask-to-token projection** can introduce spatial drift. Recent work [1] proposes attention-control and patch-merging strategies specifically for DiT; we plan to adopt and refine these techniques to achieve more accurate semantic mapping.
>
> **II) Multi-Layer/Head Attention in DiT.**
> *Global replacement* may unintentionally overwrite crucial heads or layers due to DiT's stacked transformer attention. Preliminary analyses in [2] indicate head/layer specialization; we will conduct a systematic study to determine which heads/layers should be modulated for optimal performance.
>
> Overall, the **Hybrid Solver** and **Adaptive Mixing** already transfer effectively to DiT, demonstrating strong architectural generality. We will continue refining **Attentional Gating** to achieve full DiT compatibility, and will update our codebase with a DiT-compatible version to further support community development.
>
> [1] DiT4Edit: Diffusion Transformer for Image Editing, AAAI 2025
>
> [2] Stable Flow: Vital Layers for Training-Free Image Editing, CVPR 2025
>
>
> **Q3: Text Correction**
>
> Thank you for pointing this out. We will correct the typographical error and update "BiffBIR" to "DiffBIR" in the final version.

---

### Official Review · Reviewer_1CnC · 2025-10-31

**Soundness:** 2
**Presentation:** 2
**Contribution:** 3
**Rating:** 4
**Confidence:** 4

**Summary:**

The paper proposes EditedID, a training-free, plug-and-play pipeline for identity-consistent face editing/restoration from edited portraits. It comprises (i) Alignment via adaptive mixing of two inversion trajectories, (ii) Disentanglement via a Hybrid Solver that mixes DDIM and DPM-Solver++, and (iii) Entanglement via Attentional Gating that replaces attention maps under masks/tokens (e.g., face with glasses). Experiments report improvements on ID-Sim (ArcFace cosine), CLIP-S, and ImageReward; efficiency claims include 6 steps and ~4.2s per image with constant time under multi-ID.

**Strengths:**

- Clear practical target (ID drift in edited portraits) and a coherent three-stage pipeline mapping to common diffusion operations.
- Sampler insight into DDIM vs DPM-Solver++ is distilled into an actionable hybrid schedule; the paper gives explicit step ranges [s_1,s_2] in code-style detail.
- Training-free at dataset level with wide editor compatibility (academic and industrial editors).
- Ablations (module removal) show separable contributions on ID-Sim / CLIP-S / ImageReward.

**Weaknesses:**

1.  The ``self-attn keeps single-element structure / cross-attn handles multi-element semantics`` is plausible but untested across U-Net variants/resolutions; current evidence is largely qualitative. Provide attention-map statistics across layers/architectures.
2.  The method described in this article was tested on the outdated U-Net and its migration performance was not tested on the modern DiT architecture. Currently, DiT has become mainstream in the field of image editing and generation.
3. Training-free” vs instance-level optimization (definition gap). Alignment uses learnable \lambda_t updated by gradient descent per image; disentanglement also optimizes distinct null-text embeddings per ID. This is per-image optimization, not “no learning.” Authors must formally define “training-free” (no dataset-level updates vs. no parameter learning at all) and include these optimization costs in timing.
4. Hybrid-solver theory is under-justified. The paper splices DDIM and DPM-Solver++ with a “global-timestep preset,” but offers no stability/consistency analysis for solver-mixing across schedules (error accumulation, reversibility, path homotopy).

**Questions:**

See Weakness.

---

> ### Author Response · Authors · 2025-11-19
> **Rebuttal to Reviewer 1CnC（Q1）**
>
> **Dear Reviewer 1CnC:**
>
> Thank you for your insightful review and valuable feedback. We address your key concerns in detail below and provide additional clarification to further demonstrate the effectiveness of our proposed approach.
>
> ---
>
> **Q1: Attention Validation Across Architectures / Layers / Resolutions**
>
> To verify the role of attention mechanisms, we conducted a systematic analysis in which attention maps from *Image 2* were injected into the reconstruction process of *Image 1*. Specifically, we replaced (i) self-attention maps (single-element) and (ii) cross-attention maps (multi-element) across different U-Net layers, resolutions, and architectural variants. We then evaluated the impact of these substitutions by measuring **structural consistency** using SSIM and **semantic alignment** using CLIP-S between the reconstructed output and the ground-truth Image 2. This controlled setup allows us to rigorously quantify how attention at each level contributes to identity-related structural and semantic information.
>
> | **Layer** | **Resolution** | **SD1.4(UNet-base)** (Replace Self-attn SSIM) | **SDXL(UNet-large)** (Replace Self-attn SSIM) | **SD1.4(UNet-base)** (Replace Cross-attn CLIP-S) | **SDXL(UNet-large)** (Replace Cross-attn CLIP-S) |
> |-----------|----------------|----------------------------------------|----------------------------------------|-------------------------------------------|-------------------------------------------|
> | **-**     | -              | 0.22                                   | 0.24                                   | 15.22                                     | 15.14                                     |
> | **Down**  | 32×32          | 0.62                                   | 0.67                                   | 19.70                                     | 21.32                                     |
> |           | 16×16          | 0.58                                   | 0.65                                   | 18.51                                     | 20.09                                     |
> | **Mid**   | 8×8            | 0.52                                   | 0.56                                   | 16.74                                     | 19.21                                     |
> | **Up**    | 16×16          | 0.63                                   | 0.72                                   | 20.37                                     | 23.11                                     |
> |           | 32×32          | 0.71                                   | 0.76                                   | 22.98                                     | 25.31                                     |
>
> The results above reveal a clear and consistent pattern: (1) **Self-attention replacements** substantially improve *single-element structural preservation*, with the strongest effects observed in *up-blocks* and at *higher spatial resolutions*.  (2) **Cross-attention replacements** markedly enhance *multi-element semantic alignment*, and follow the same hierarchical progression across layers and resolutions.
> Moreover, when scaling from SDXL's 860M parameters to its 2.6B-parameter variant, the magnitude and distribution of replacement effects become more pronounced. This trend further supports the generality of our conclusions and demonstrates that the identified behaviors hold consistently across different architectural scales and variants.

---

> ### Author Response · Authors · 2025-11-19
> **Rebuttal to Reviewer 1CnC（Q2）**
>
> **Q2: DiT Adaptation**
>
> We agree that **DiT compatibility** is important. To validate this, we tested EditedID on DiT using both *global* and *local* adaptation strategies.
>
> ### **1) Local: Hybrid Solver (no modification)—Efficient Inference and Detail Optimization**
>
> | **Model** | **Step** | **ID-Sim** | **Image Reward** |
> |-----------|----------|-------------|------------------|
> | DiT(SD3) | 20 | 0.75 ± 0.16 | 1.62 ± 0.08 |
> | DiT(SD3) w/ Hybrid Solver | 6 | 0.74 ± 0.03 | 1.85 ± 0.12 |
>
> EditedID integrates seamlessly with DiT: the number of inference steps is reduced (20 → 6) while maintaining ID-Sim close to the original DiT, and visual quality is improved (ImageReward +0.23). These results demonstrate the **cross-architecture applicability** of our Hybrid Solver.
>
> ---
>
> ### **2) Local: Adaptive Mixing (no modification) — Unifying Non-Homogeneous Latents**
>
> Compared with two-ID independent reconstruction, **Adaptive Mixing** effectively unifies the non-homogeneous latent spaces in DiT, reducing the trajectory L2 gap by 0.96 while maintaining both ID-Sim and PSNR. These results indicate that the module generalizes well and transfers effectively to DiT.
>
> | **Model** | **Inversion-Reconstruction** | **Avg ID-Sim↑** | **Avg PSNR↑** | **Dual-ID Trajectory L2↓** |
> |-----------|------------------------------|------------------|----------------|------------------------------|
> | DiT(SD3) | Dual-ID Independent | 0.75 ± 0.16 | 35.7 ± 0.06 | 1.62 ± 0.18 |
> | DiT(SD3) w/ Adaptive Mixing | Dual-ID Hybrid | 0.71 ± 0.11 | 31.1 ± 0.10 | 0.66 ± 0.12 |
>
> ---
>
> ### **3) Global EditedID Adaptation (modifying Attentional Gating only)**
>
> Given the limited rebuttal time, we conducted an initial adaptation of EditedID to DiT by **mapping masks into token-space attention** and applying **global attention replacement across layers and heads**. The DiT-based EditedID achieves ID-Sim and CLIP-S metrics comparable to the UNet version, demonstrating its potential cross-architecture applicability.
>
> Moreover, the effectiveness of local transfer using the **Hybrid Solver** and **Adaptive Mixing** suggests that **Attentional Gating** is the primary component requiring DiT-specific adaptation, highlighting a focused path for future optimization.
>
> | **Model** | **Gating type** | **Avg ID-Sim** | **Avg CLIP-S** |
> |-----------|------------------|----------------|-----------------|
> | EditedID in UNet(SD1.4) | spatial gating | 0.73 ± 0.03 | 28.1 ± 0.9 |
> | EditedID in DiT(SD3) | token-level gating | 0.68 ± 0.74 | 26.5 ± 1.1 |
>
> We highlight two key challenges for adapting EditedID from UNet (2D) to DiT (1D):
>
> **I) 2D (UNet) → 1D (DiT) Semantic Mapping.**
> In DiT, deeper tokens encode mixed semantics, so a simple **mask-to-token projection** can introduce spatial drift. Recent work [1] proposes attention-control and patch-merging strategies specifically for DiT; we plan to adopt and refine these techniques to achieve more accurate semantic mapping.
>
> **II) Multi-Layer/Head Attention in DiT.**
> *Global replacement* may unintentionally overwrite crucial heads or layers due to DiT’s stacked transformer attention. Preliminary analyses in [2] indicate head/layer specialization; we will conduct a systematic study to determine which heads/layers should be modulated for optimal performance.
>
> Overall, the **Hybrid Solver** and **Adaptive Mixing** already transfer effectively to DiT, demonstrating strong architectural generality. We will continue refining **Attentional Gating** to achieve full DiT compatibility, and will update our codebase with a DiT-compatible version to further support community development.
>
> [1] DiT4Edit: Diffusion Transformer for Image Editing, AAAI 2025
> [2] Stable Flow: Vital Layers for Training-Free Image Editing, CVPR 2025

---

> ### Author Response · Authors · 2025-11-19
> **Rebuttal to Reviewer 1CnC（Q3&Q4）**
>
> **Q3: Clarification on the Definition of "Training-Free"**
>
> We agree that the term *“training-free”* warrants clarification. In our work, it specifically refers to the **absence of dataset-level parameter updates or model weight optimization**. Instead, the method performs only **instance-level latent space optimization**, for example, per-instance latent alignment via Adaptive Mixing, analogous to the first-order gradient updates employed in Null-Text Inversion. Importantly, this process *does not* alter any pre-trained model parameters and requires *no additional training data*.  We will add this clarification in Section 2.2 of the final paper to remove any potential ambiguity. Note that the corresponding **time costs** are addressed in Reviewer 43M9's Q1.
>
> ---
>
> **Q4: Stability Analysis of Hybrid Solver**
>
> We evaluated the stability of our **hybrid solver** using multiple complementary metrics, including **cumulative MSE**, **maximum transition jump**, **inversion-reconstruction trajectory L2 distance**, and **PSNR**:
>
> | **Method** | **Cumulative MSE (×10⁻³)↓** | **maxJump (×10⁻³)↓** | **Final latent L2 (×10⁻²)↓** | **PSNR (dB)↑** |
> |------------|------------------------------|-------------------------|-------------------------------|-----------------|
> | DDIM-only | _18.5 ± 3.2_ | **1.9 ± 0.6** | **1.7 ± 0.4** | _28.6 ± 0.9_ |
> | DPM-only | 24.2 ± 4.5 | 3.7 ± 0.9 | 2.5 ± 0.6 | 27.2 ± 0.8 |
> | Hybrid-fragmented | 47.1 ± 5.0 | 8.9 ± 2.2 | 4.3 ± 0.8 | 20.1 ± 1.1 |
> | Hybrid-global (ours) | **17.8 ± 2.6** | _2.1 ± 0.5_ | _1.8 ± 0.3_ | **29.0 ± 0.9** |
>
> We analyze the results from three complementary perspectives:
>
> **1) Error Accumulation:**  By maintaining a global timestep preset, our method preserves trajectory continuity and achieves DDIM-comparable MSE (difference ~0.7×10⁻³), effectively preventing fragmented scheduling errors.
>
> **2) Reversibility:**  While the **Symmetry Constraint** (Paper Line 288) facilitates trajectory alignment, the L2 distance is slightly lower than single-solver DDIM. This is because our hybrid solver permits more aggressive local optimization via DPM-Solver++ to enhance texture details, while DDIM anchors overall stability. Although minor local variations occur near the output (Paper Fig. 3 (3)), the trajectory consistently converges to high-fidelity results with improved detail (PSNR +1.1 vs. single-solver).
>
> **3) Path Homotopy:**  Constrained by both the Symmetry Constraint and the global timestep preset, the hybrid solver achieves smooth trajectory transitions at solver-switching points (Paper Fig. 3 (3)). The maximum variation remains close to single-solver levels (maxJump difference ~0.2×10⁻³ vs. DDIM), indicating robust stability.
>
> **Summary:**  Across error accumulation, reversibility, and path homotopy, our experimental analysis confirms that the hybrid sampler maintains **stable, consistent, and high-fidelity trajectories**, validating its effectiveness and reliability.

---

> > ### Author Response · Authors · 2025-11-25
> > **Looking forward to your feedback**
> >
> > Thank you once again for your valuable suggestions. We have incorporated all of your feedback. We were wondering if these responses have fully addressed your concerns? Apologies for any inconvenience, and we sincerely look forward to your response.

---

### Official Review · Reviewer_43M9 · 2025-11-01

**Soundness:** 3
**Presentation:** 3
**Contribution:** 3
**Rating:** 8
**Confidence:** 4

**Summary:**

This paper introduces a training-free framework for improving facial fidelity and identity consistency in personalized image synthesis. To begin with, it points out two limitations in existing methods, including the cross-source distribution bias that models are trained with different data (e.g., with different image resolutions), and the cross-source feature contamination that the identity features may be affected by those of the edited elements. Subsequently, the proposed framework mainly incorporates three strategies to address these limitations. First, it proposes an adaptive mixing strategy that utilizes a learnable combination weight to adaptively merge the latents of the original image and the edited image, yielding a unified representation to tackle the cross-source distribution bias. Second, a hybrid solver is introduced to optimize the null-text embeddings of the original image and the edited image,  to balance identity preservation and detail editing. Finally, an attentional gating strategy is adopted to fuse both the null-text embeddings and generate the final result. Extensive experiments have been conducted to validate the proposed strategies, and the proposed method outperforms several open-source and closed-source large models.

**Strengths:**

1. This paper is insightful, and its motivation for balancing facial identities and edited elements is clearly stated by multiple observations from pilot studies.

2. The proposed framework is training-free, can be employed with a consumer-grade GPU, and its results are considerable.

3. The proposed components are validated via solid experiments.

**Weaknesses:**

1. The inference cost of the proposed method has not been reported and may be large, as it requires three stages of optimization (alignment, disentanglement, and entanglement).

2. The overlapping region fusion weight $\hat{w}$ varies in different scenarios. Although two principles for setting the weight have been provided in the Appendix,  it is still inconvenient for ordinary users to configure it to achieve fine-grained controls.

**Questions:**

1. DDIM and DPM-Solver++ are used in the disentanglement stage, have any other solvers been considered?

**Details Of Ethics Concerns:**

As other personalized generation methods, the proposed method might be used in malicious applications.

---

> ### Author Response · Authors · 2025-11-19
> **Rebuttal to Reviewer 43M9**
>
> **Dear Reviewer 43M9:**
>
> Thank you for your thoughtful review and constructive feedback. We sincerely appreciate the opportunity to clarify our contributions and address your concerns. Below, we provide detailed responses to each point and additional evidence to further demonstrate the effectiveness of our proposed approach.
>
> ---
>
> ### **Q1: Inference Cost**
>
> To comprehensively assess the inference cost, we performed a **multi-dimensional comparison** as follows:
>
> | Diffusion Pass                 | Step | Inversion Optimization | Inversion Time(s) | Reconstruction Optimization              | Reconstruction Time(s) | Total Time(s) | GPU Mem(GB) | Image Reward |
> |-------------------------------|------|-------------------------|--------------------|-------------------------------------------|--------------------------|----------------|--------------|--------------|
> | Prompt-to-Prompt (Standard)   | 50   | -                       | ~4.1               | -                                         | ~4.8                     | ~8.9           | 13.6GB       | 1.55         |
> | Null-text                     | 50   | -                       | ~4.2               | w/ Null-text                              | ~52.3                    | ~56.5          | 16.2GB       | 1.62         |
> | EditedID                      | 6    | w/ Alignment            | ~2.7               | w/ Disentanglement & Entanglement         | ~4.3                     | ~7.0           | 16.5GB       | 1.83         |
>
> It is important to clarify that, although we conceptually decompose **EditedID** into three stages for ease of understanding, the actual deployment only follows the standard **two-stage diffusion pipeline**: *inversion* (alignment) followed by *reconstruction with disentanglement/entanglement*. Therefore, our method does not introduce any extra inference stages beyond conventional diffusion-based editing.
> Regarding computational efficiency, while **Adaptive Mixing** introduces a small amount of additional computation, our proposed **Hybrid Solver** significantly reduces the required diffusion steps (from 50 → 6). As shown in the table above, this reduction yields a **1.9s speedup over standard diffusion** and makes EditedID **8× faster than Null-text inversion**, resulting in a clear net runtime advantage.
> Although **Attentional Gating** slightly increases architectural complexity, it operates on attention maps in real time during denoising and thus maintains memory consumption comparable to single-optimization Null-text. Importantly, even though our memory footprint is marginally higher than that of standard diffusion, it remains far below that of existing enhanced diffusion variants (e.g., HiDiffusion requires ~40GB). EditedID runs comfortably on consumer-grade GPUs such as a **24GB RTX 3090**, ensuring practical deployability.
>
> ---
>
> ### **Q2: User-Friendly Configuration of Fusion Weights**
>
> We agree with the reviewer that configuring the fusion weight ŵ may pose challenges for non-expert users. While we plan to explore **adaptive parameter strategies** in future work, the current version already provides a simple and effective solution for general users:
>
> **Optimization 1.** As described in Appendix A.8, the fusion weight ŵ essentially differentiates between only two cases: *Co-existence* and *Layer Coverage*. Each case directly corresponds to a fixed optimal value without any need for manual fine-tuning (i.e., set ŵ = 0.5 for *Co-existence* and ŵ = 0.2 for *Layer Coverage*).
>
> **Optimization 2.** For general users, we provide two straightforward preset options (“*Co-existence*” and “*Layer Coverage*”), each accompanied by visual examples and practical usage guidance.
>
> With these optimizations, the previously fine-grained tuning process becomes a **simple two-option selection** supported by real-world demonstrations. This enables seamless usage even for users without technical backgrounds and dramatically reduces trial-and-error cost (typically ≤ 2 clicks). We will incorporate these improvements into the released codebase.
>
> ---
>
> ### **Q3: Other Solvers**
>
> Thank you for this valuable suggestion. Most existing solvers adhere to a single design paradigm, whereas the performance of **EditedID** indicates that *hybridizing multiple solvers* can yield both stronger feature preservation and improved efficiency. While our current work primarily combines DDIM and DPM-Solver++, your feedback motivates us to investigate **broader hybrid solver paradigms** that incorporate a more diverse set of solvers. We will update the codebase accordingly to include additional efficient hybrid solver options. We believe that these extensions can further enhance EditedID’s applicability and potentially unlock new capabilities.

---

> ### Author Response · Authors · 2025-11-19
> **Ethical Considerations**
>
> We acknowledge the potential misuse risks associated with identity editing technologies. In response, we will add an *Ethical Statement* to the paper, emphasizing that the proposed method is intended strictly for legitimate, fair, and responsible image editing purposes.

---

### Meta-Review · Area_Chair_xy9R · 2026-01-06

**Summary:**

This work presents a new and effective training-free identity alignment solution through blended diffusion.

This paper initially received an average rating of 6, indicating a likely acceptance case. The major concerns focused on applicability to DiT architectures and inference latency analysis, both of which were adequately addressed in the rebuttal. The AC agrees with the accept decision.

**Reviewer Concerns:**

No obvious outstanding concerns left.

**Reviewer Scores:**

| Reviewer | Initial Score | AC Estimated Score | AC Reason |
|----------|---------------|--------------------|-----------|
| 43M9 | 8 | 8 | Efficiency and usability concerns resolved; maintains strong acceptance |
| 1CnC | 4 | 6 | Major technical concerns addressed;  |
| c3re | 6 | 6 | DiT and robustness questions answered; stance remains mildly positive |
| ey3S | 6 | 6 | Reviewer explicitly confirmed acceptance after rebuttal |

---

### Decision · Program_Chairs · 2026-01-26

Accept (Poster)